# Double and triple thermodynamic mutant cycles reveal the basis for specific MsbA-lipid interactions

Jixing Lyu[1], Tianqi Zhang[1], Michael T Marty[2], David Clemmer[3], David H Russell[1], Arthur Laganowsky[1]*

[1]Department of Chemistry, Texas A&M University, College Station, United States; [2]Department of Chemistry and Biochemistry and Bio5 Institute, The University of Arizona, Tucson, United States; [3]Department of Chemistry, Indiana University, Bloomington, United States

*For correspondence: alaganowsky@chem.tamu.edu

Competing interest: The authors declare that no competing interests exist.

**Abstract** Structural and functional studies of the ATP-binding cassette transporter MsbA have revealed two distinct lipopolysaccharide (LPS) binding sites: one located in the central cavity and the other at a membrane-facing, exterior site. Although these binding sites are known to be important for MsbA function, the thermodynamic basis for these specific MsbA-LPS interactions is not well understood. Here, we use native mass spectrometry to determine the thermodynamics of MsbA interacting with the LPS-precursor 3-deoxy-D-*manno*-oct-2-ulosonic acid $(Kdo)_2$-lipid A (KDL). The binding of KDL is solely driven by entropy, despite the transporter adopting an inward-facing conformation or trapped in an outward-facing conformation with adenosine 5'-diphosphate and vanadate. An extension of the mutant cycle approach is employed to probe basic residues that interact with KDL. We find the molecular recognition of KDL is driven by a positive coupling entropy (as large as −100 kJ/mol at 298 K) that outweighs unfavorable coupling enthalpy. These findings indicate that alterations in solvent reorganization and conformational entropy can contribute significantly to the free energy of protein-lipid association. The results presented herein showcase the advantage of native MS to obtain thermodynamic insight into protein-lipid interactions that would otherwise be intractable using traditional approaches, and this enabling technology will be instrumental in the life sciences and drug discovery.

## eLife assessment

This is an **important** biophysical study combining native mass spectrometry with mutant cycles to estimate the thermodynamic components of lipid A binding to the ABC transporter MsbA. **Solid** evidence supports the binding energies for lipid-protein interactions to MsbA using this approach, which could be later applied to other membrane proteins in general.

## Introduction

Most Gram-negative bacteria contain outer membrane lipopolysaccharide (LPS) that is crucial for maintaining structural integrity and protection from toxins and antibiotics (*Simpson and Trent, 2019*; *Raetz and Whitfield, 2002*; *Raetz et al., 2007*). The **A**TP-**B**inding **C**assette (ABC) transporter MsbA flips an LPS-precursor, lipooligosaccharide (LOS), from the cytosolic leaflet to the periplasmic leaflet of inner membrane, a process powered by the hydrolysis of adenosine triphosphate (ATP). MsbA functions as a homodimer and each subunit consists of a soluble nucleotide-binding domain (NBD) and a transmembrane domain containing six transmembrane helices (*Rees et al., 2009*). The proposed

mechanism of MsbA-mediated LOS transportation involves the binding of LOS to the interior binding site and a conformational change from an inward-facing conformation (IF) to an outward-facing conformation (OF).

Like other ABC transporters, the ATPase activity of MsbA can be stimulated in the presence of different substrates, particularly hexaacylated lipid A species. (*Doerrler and Raetz, 2002*; *Eckford and Sharom, 2008*; *Siarheyeva and Sharom, 2009*) Recent studies have illuminated the location and importance of several LOS binding sites on MsbA (*Mi et al., 2017*; *Ho et al., 2018*; *Lyu et al., 2022*; *Padayatti et al., 2019*). The interior binding site is located in the inner cavity, and mutations (R78A, R148A and K299A) engineered to disrupt binding at this site abolish lipid-stimulated ATPase activity and adversely affect cell growth. (*Mi et al., 2017*; *Guo et al., 2021*) More recently, the LPS-precursor 3-deoxy-D-manno-oct-2-ulosonic (Kdo)2-lipid A (KDL) was found to bind to an elusive exterior site on MsbA trapped in an OF conformation with adenosine 5'-diphosphate and vanadate (*Doerrler and Raetz, 2002*; *Eckford and Sharom, 2008*; *Siarheyeva and Sharom, 2009*). Similarly, introducing mutations to disrupt binding at the exterior site also abolishes lipid-induced stimulation of ATPase activity (*Lyu et al., 2022*).

In 1984, Fersht and colleagues introduced the biochemistry community to the application of double mutant cycles as means to quantify the strength of intramolecular and intermolecular interactions. (*Carter et al., 1984*) The method has proven to be highly effective in examining pairwise interactions, as demonstrated by its notable application in determining the spatial orientation of potassium channel residues in relation to high-affinity toxin binding (*Hidalgo and MacKinnon, 1995*). More generally, the technique has been used to measure the strength and coupling for residues in protein-protein complexes, protein-ligand complexes, and stability of secondary structure. (*Carter et al., 1984*; *Hidalgo and MacKinnon, 1995*; *Horovitz, 1996*; *Pagano et al., 2021*; *Horovitz et al., 2019*; *Cockroft and Hunter, 2007*; *Otzen and Fersht, 1999*; *Schreiber and Fersht, 1995*; *Thomas-Tran and Du Bois, 2016*) In general, mutant cycles analysis involves measuring the changes in Gibbs free energy for the wild-type protein (P), two single point mutations (PX and PY) and the double mutant protein (PXY) for a given process, such as for protein-protein interactions (for review see *Horovitz, 1996*). If residue X and Y are independent of each other, then the Gibbs free energy associated with the double mutant protein will be equal to the sum of changes in Gibbs free energy due to the single mutations relative to the wild-type protein. However, if the Gibbs free energy associated with the structural and functional properties of the double mutant protein differs from the sum of single mutant proteins, then the two residues are energetically coupled or co-operative. The coupling free energy ($\Delta\Delta G_{int}$) is the energy difference between double mutant and two single mutant proteins (see Materials and methods). The $\Delta\Delta G_{int}$ values for pairwise interactions in proteins has revealed the contributions of salt bridges (4–20 kJ/mol), aromatic-aromatic interactions (4 kJ/mol), and charge-aromatic interactions (4 kJ/mol) to protein stability. (*Horovitz, 1996*; *Luisi et al., 2003*; *Serrano et al., 1991*) Prior work on mutant cycles often employed traditional approaches, but such approaches overlook contributions from conformational changes of the reactants as well as potential changes in the hydration of the complex, including the reacting ligand and the solvent (*Pagano et al., 2021*).

To demonstrate the utility of the mutant cycle approach, we highlight two well-known examples. First, the high-affinity interaction between barnase (an extracellular RNase of *Bacillus amyloliquefaciens*) and barstar (inhibitor of barnase) has been extensively studied by double mutant cycles. (*Schreiber and Fersht, 1995*) For example, pairwise interactions between residues that are less than seven angstrom in distance (based on crystal structures) have been shown to be co-operative. These interactions were shown to be important for stability of the barnase-barstar complex with coupling energies reaching as high as 7 kcal/mol. Another classical example involves the application of mutant cycles to guide docking and spatial arrangement of a high-affinity peptide inhibitor (scorpion toxin) binding to the Shaker potassium channel (*Hidalgo and MacKinnon, 1995*). Of the pairwise interactions that underwent mutant cycle analysis, one pair (R24 from toxin and D431 from channel) in particular displayed an extraordinary coupling energy of 17 kJ/mol. This result indicates the two residues interact in the complex. Despite the absence of a structure of toxin-potassium channel complex, results from the mutant cycle analysis provided a strong constraint positioning the toxin relative to the potassium channel pore-forming region. In summary, these studies demonstrate how mutant cycle analysis can be used to determine the energetics of pairwise interactions, which is important

for understanding how these molecular interactions contribute to the overall stability of proteins in complex with other molecules, such as ligands and other proteins.

Native mass spectrometry (MS) is well suited to characterize the interactions between protein and other molecules, especially for membrane proteins (*Bolla et al., 2019*; *Robinson, 2017*; *Tamara et al., 2022*). The technique is capable of maintaining non-covalent interactions and native-like structure in the gas phase (*Ruotolo et al., 2005*; *Laganowsky et al., 2014*), essential for studying biochemical interactions with small molecules, such as the binding of drugs, lipids, and nucleotides (*Laganowsky et al., 2014*; *Allison et al., 2015*; *Barrera et al., 2008*; *Campuzano et al., 2019*; *Gupta et al., 2017*; *Marcoux et al., 2013*; *Yen et al., 2018*; *Zhou et al., 2011*). In combination with a variable temperature nano electrospray ionization device, native MS has determined the thermodynamics for protein-protein and protein-ligand interactions (*Daneshfar et al., 2004*; *Deng et al., 2013*; *Raab et al., 2020*; *Walker et al., 2023*; *Qiao et al., 2021*; *McCabe et al., 2021*). For example, the molecular interaction between the signaling lipid 4,5-bisphosphate phosphatidylinositol and Kir3.2 is dominated by a large, favorable change in entropy (*Qiao et al., 2021*). Recently, native MS has been combined with mutant cycles analysis to determine the energetic contribution of pairwise inter-protein interactions for a soluble protein complex (*Sokolovski et al., 2017*; *Cveticanin et al., 2018*). Notably, the coupling energies determined by native MS and isothermal calorimetry are in agreement (*Sokolovski et al., 2017*). Mutant cycle analysis is also being used to study cardiolipin binding to sites on AqpZ with native MS (*Jayasekera et al., 2023*).

Traditional mutant cycles focus on pairwise interactions, such as two interacting residues in a protein complex (*Serrano et al., 1990*). Single- and double-point mutations along with characterizing their impact on protein stability/assembly enable assessment of the energetic contribution for the pairwise interaction. If the two residues are independent (non-co-operative) then the change in free energy will be equal to the sum of the two single mutations. In contrast, if the two residues are dependent on each other, then the coupling energy is a measure of their co-operativity. Although mutant cycles are often applied to protein-protein interactions, here we extend mutant cycle principles to study membrane protein-lipid interactions. It is established that MsbA has two high-affinity binding sites for the LPS-precursor KDL. Here, we examine each site independently followed by simultaneously probing both KDL sites. At present, there is limited availability of synthetic KDL derivatives, limiting this study to focus on residues that interact with KDL, such as basic residues coordinating the conserved phosphoglucosamine (P-GlcN) of KDL. Despite the limitation of commercially available KDL derivatives, the studies below demonstrate how residues energetically contribute to specific binding, providing insight into the driving forces underlying essential membrane protein-lipid interactions. Recently, we reported results using native MS that reveal conformation-dependent lipid binding affinities to MsbA (*Lyu et al., 2022*). As these measurements were performed at a single temperature, we set out to perform a more detailed thermodynamic analysis to better understand the molecular driving forces that underpin specific MsbA-lipid interactions. Here, we report binding thermodynamics ($\Delta H$, $\Delta S$, and $\Delta G$) for KDL binding to MsbA in IF and OF conformations. These results reveal the unique thermodynamic contributions of MsbA residues that engage KDL. We also report coupling energetics ($\Delta\Delta G_{int}$) for pairwise interactions, including, for the first time, the contributions from coupling enthalpy ($\Delta\Delta H_{int}$) and coupling entropy ($\Delta(-T\Delta S_{int})$), providing rich molecular insight into specific protein-lipid interactions.

## Results
### MsbA residues selected for mutant cycles analyses

MsbA is known to bind KDL either in the inner cavity or at the two exterior sites (*Figure 1*). For both sites, a series of conserved arginine and lysine residues form specific interactions with the headgroup of KDL. To perform mutant cycles analysis, we introduced single mutations into MsbA to target KDL binding to the interior (MsbA[R78A] and MsbA[R299A]) and exterior (MsbA[R188A], MsbA[R238A], and MsbA[K243A]) sites. More specifically, R78 coordinates one of the characteristic phosphoglucosamine (P-GlcN) substituents of KDL whereas K299 interacts with a carboxylic acid group in the headgroup of KDL. The two P-GlcN constituents of LOS are coordinated by R238 and R188 +K243, respectively. R188 also forms an additional hydrogen bond with the headgroup of KDL. In addition, we prepared double and triple mutants of MsbA for the various residues that were selected for mutagenesis.

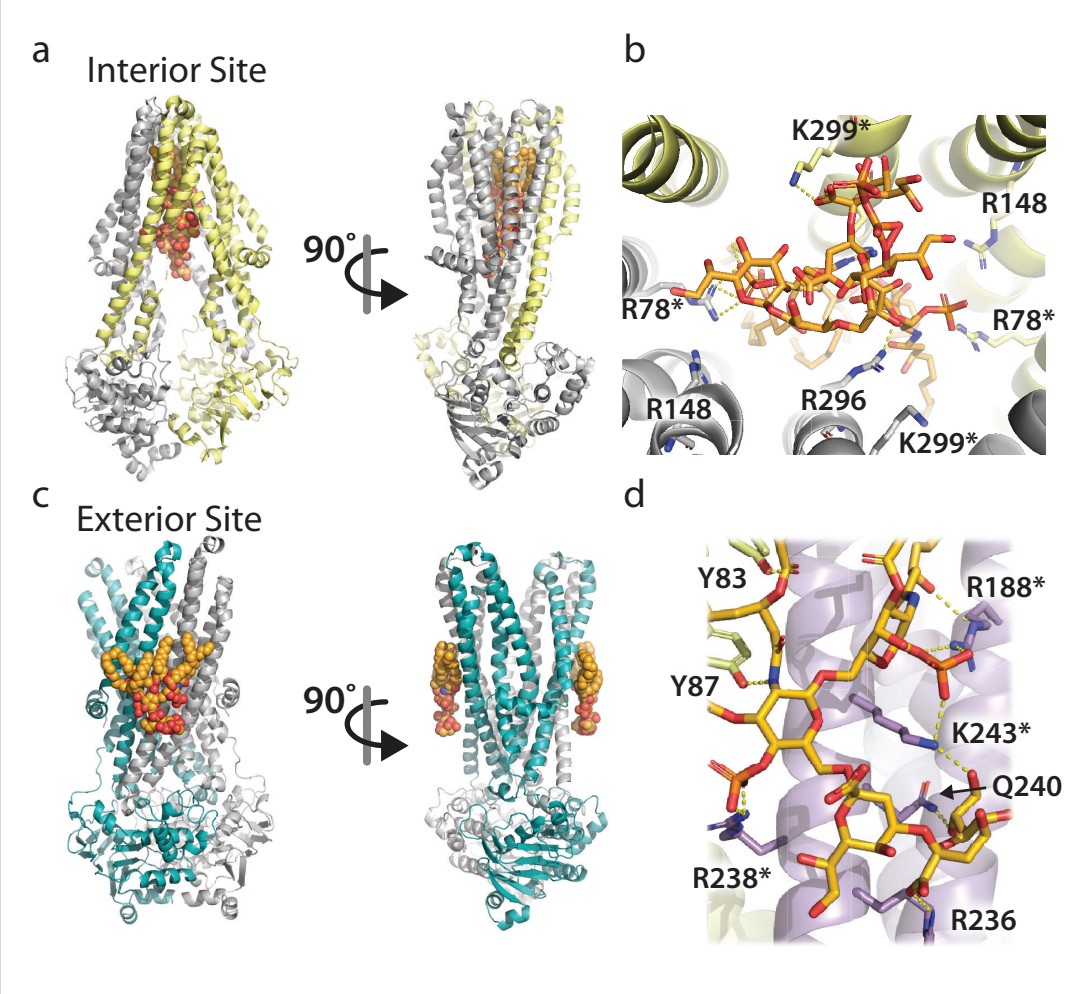

**Figure 1.** The two distinct LPS binding sites of MsbA and their molecular interactions. (**a**) Two views of LPS bound to the interior site or central cavity of MsbA. The protein shown is also bound to the inhibitor G907 (PDB 6BPL) (*Ho et al., 2018*). The protein and lipid are shown in cartoon and stick representation, respectively. (**b**) Molecular details of the residues interacting with LPS at the interior site. Bonds are shown as dashed yellow lines along with residue labels. (**c**) Two views of the KDL molecules bound to the two exterior binding sites of MsbA that are symmetrically related (PDB 8DMM) (*Lyu et al., 2022*). Shown as described in panel A. (**d**) Molecular view of KDL bound to MsbA and shown as described in panel B. The asterisk denotes residues selected for mutant cycle analysis.

## Thermodynamics of MsbA-KDL interactions

We performed titrations to determine the equilibrium binding affinity for MsbA-KDL interactions at four different temperatures (288, 293, 298, and 303 K; *Figure 2*, *Figure 2—figure supplements 1–3*). These studies used optimized samples of MsbA that do not contain any co-purified LPS (for details see *Lyu et al., 2022*). The transporter was stable at the selected temperatures. For example, binding of KDL to MsbA was enhanced at higher temperatures (*Figure 2a*, *Figure 2—figure supplements 1–3*), indicating a favorable entropy for the interaction. For a given temperature, the mass spectra from the titration series were deconvoluted and equilibrium dissociation constants ($K_D$) were determined for MsbA binding up to three KDL molecules (*Figure 2b*, *Figure 2—figure supplements 1–3* and *Table 1*). It is important to note that, unlike traditional approaches that struggle to distinguish between free protein from that bound to ligand, (*Jarmoskaite et al., 2020*) native MS can resolve different ligand bound states, including the free concentration of protein and free concentration of ligand(s), in a single mass spectrum (*Daneshfar et al., 2004*; *Cong et al., 2016*; *Cong et al., 2017*; *Patrick et al., 2018*). Notably, the native MS approach has been cross validated using isothermal calorimetry and surface plasmon resonance (*Daneshfar et al., 2004*; *Cong et al., 2016*; *Cong et al., 2017*). Interestingly, van' t Hoff analysis showed a non-linear trend for three KDL binding reactions

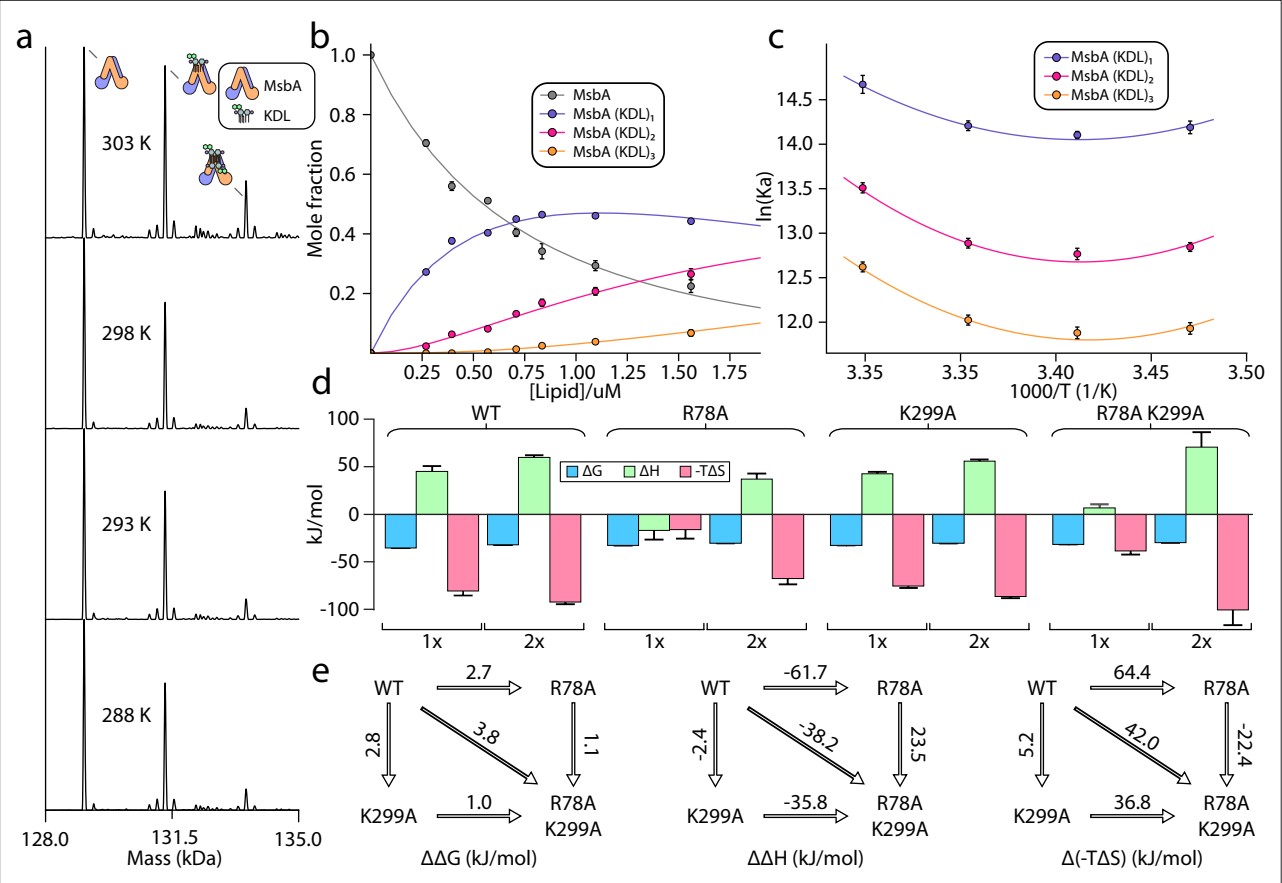

**Figure 2.** Thermodynamics of KDL binding at the interior site to wild-type and mutant MsbA. (**a**) Representative deconvoluted native mass spectra of 0.39 µM wild-type MsbA in $C_{10}E_5$ and in the presence of 0.6 µM KDL recorded at different solution temperatures. (**b**) Plot of mole fraction of MsbA $(KDL)_{0-3}$ determined from titration of KDL (dots) at 298 K and resulting fit from a sequential ligand binding model (solid line, $R^2$=0.99). (**c**) van' t Hoff plot for MsbA$(KDL)_{1-3}$ and resulting fit of a nonlinear van' t Hoff equation. (**d**) Thermodynamics for MsbA and mutants (MsbA$^{R78A}$, MsbA$^{K299A}$ and MsbA$^{R78A,K299A}$) binding KDL at 298 K. (**e**) Mutant cycles for MsbA and mutants with (from left to right) ΔΔG (mutant minus wild-type), ΔΔH and Δ(-TΔS) values indicated over the respective arrows. Shown are values at 298 K. Reported are the average and standard deviation from repeated measurements (n = 3).

The online version of this article includes the following figure supplement(s) for figure 2:

**Figure supplement 1.** Representative native mass spectra of wild-type and mutant MsbA in the presence of 0.8 µM KDL.

**Figure supplement 2.** Determination of equilibrium dissociation constants ($K_D$) for KDL binding wild-type and mutant MsbA.

**Figure supplement 3.** Determination of thermodynamic parameters for KDL binding wild-type and mutant MsbA.

(*Figure 2c*, *Figure 2—figure supplements 1–3*), indicating that over the selected temperature range, heat capacity is not constant (*Prabhu and Sharp, 2005*). The nonlinear form of the van't Hoff equation enabled us to determine the ΔH and change in heat capacity ($ΔC_p$) at a reference temperature of 298 K (*Figure 2c–d*, *Figure 2—figure supplements 1–3*). In this case, ΔG was calculated directly from $K_D$ values, and entropy (ΔS) was back calculated using both ΔH and ΔG. ΔG values for binding $KDL_{1-2}$ range from –32.0±0.1 to -35.2±0.1 kJ/mol. The binding reaction has a positive $ΔC_p$ that alters the thermodynamic parameters at different temperatures. At the lowest temperature, KDL binding is driven by favorable enthalpy (–36±12 to -43±7 kJ/mol) with a small entropically penalty (-TΔS, 2±11–12±7 kJ/mol at 288 K). In contrast, KDL binding at higher temperatures displays a large, favorable entropy (-TΔS, –123±12 to -146±7 kJ/mol at 303 K) that compensates a large enthalpic barrier (86±12–112±7 kJ/mol). These results highlight the role of entropy in KDL binding to MsbA that may stem from solvent reorganization.

## KDL binding to the interior binding site of MsbA

We next determined the thermodynamics of KDL binding to MsbA containing single and double mutations at the interior binding site (*Figure 2d*, *Figure 2—figure supplements 1–3*). R78 of each

**Table 1.** Equilibrium dissociation constants ($K_D$) for KDL binding MsbA at various temperatures.
Reported are the mean and standard deviation (n=3).

| | Temperature(K) | $K_{D1}$(μM) | $K_{D2}$(μM) | $K_{D3}$(μM) | $R^{2*}$ | $X^{2*}$ |
|---|---|---|---|---|---|---|
| | 288 | 0.69±0.06 | 2.64±0.16 | 6.60±0.54 | 0.99 | 0.01 |
| | 293 | 0.75±0.04 | 2.86±0.23 | 6.94±0.56 | 0.99 | 0.01 |
| | 298 | 0.68±0.05 | 2.53±0.17 | 6.01±0.40 | 0.99 | 0.01 |
| WT | 303 | 0.43±0.05 | 1.36±0.10 | 3.30±0.23 | 0.96 | 0.06 |
| | 288 | 1.91±0.17 | 6.18±0.80 | | 0.98 | 0.04 |
| | 293 | 1.87±0.18 | 6.18±0.37 | | 0.98 | 0.05 |
| | 298 | 2.00±0.20 | 5.23±0.47 | | 0.98 | 0.04 |
| R78A | 303 | 7.68±0.47 | 5.72±0.21 | | 1 | 0 |
| | 288 | 2.21±0.17 | 11.01±1.95 | | 0.99 | 0.03 |
| | 293 | 2.28±0.12 | 11.62±2.17 | | 0.99 | 0.03 |
| | 298 | 1.98±0.07 | 8.92±0.73 | | 0.99 | 0.03 |
| R188A | 303 | 1.32±0.05 | 4.37±0.13 | | 0.95 | 0.12 |
| | 288 | 2.48±0.12 | 4.74±0.45 | | 0.99 | 0.02 |
| | 293 | 3.55±0.10 | 6.21±0.40 | | 1 | 0.01 |
| | 298 | 4.70±0.07 | 9.66±0.82 | | 1 | 0.01 |
| R238A | 303 | 4.71±0.20 | 9.94±0.40 | | 1 | 0.02 |
| | 288 | 1.24±0.07 | 7.28±0.78 | | 0.99 | 0.03 |
| | 293 | 1.41±0.09 | 7.38±0.53 | | 0.99 | 0.02 |
| | 298 | 1.28±0.05 | 6.65±0.50 | | 0.99 | 0.02 |
| K243A | 303 | 0.74±0.07 | 3.73±0.32 | | 0.97 | 0.06 |
| | 288 | 2.32±0.06 | 6.07±0.33 | 17.54±6.77 | 1 | 0.01 |
| | 293 | 2.35±0.07 | 5.75±0.07 | 16.03±3.18 | 0.99 | 0.02 |
| | 298 | 2.12±0.09 | 5.07±0.12 | 14.01±1.59 | 0.99 | 0.02 |
| K299A | 303 | 1.38±0.01 | 2.91±0.02 | 5.91±0.55 | 0.97 | 0.05 |
| | 288 | 2.56±0.17 | 7.77±0.11 | | 0.98 | 0.06 |
| | 293 | 3.08±0.13 | 8.92±0.39 | | 0.99 | 0.03 |
| | 298 | 3.17±0.17 | 6.78±0.82 | | 0.99 | 0.02 |
| R78A K299A | 303 | 6.91±0.66 | 11.96±2.83 | | 0.99 | 0.03 |
| | 288 | 5.55±0.87 | | | 0.99 | 0.02 |
| | 293 | 8.51±0.67 | | | 1 | 0.01 |
| | 298 | 13.01±0.43 | | | 1 | 0 |
| R188A R238A | 303 | 11.14±0.23 | | | 1 | 0 |
| | 288 | 1.06±0.02 | 5.84±0.29 | | 0.99 | 0.01 |
| | 293 | 1.07±0.02 | 6.07±0.13 | | 0.99 | 0.01 |
| | 298 | 1.05±0.02 | 5.82±0.06 | | 0.99 | 0.02 |
| R188A K243A | 303 | 0.75±0.01 | 4.35±0.17 | | 0.99 | 0.03 |

*Table 1 continued on next page*

*Table 1 continued*

| | Temperature(K) | $K_{D1}(\mu M)$ | $K_{D2}(\mu M)$ | $K_{D3}(\mu M)$ | $R^{2*}$ | $X^{2*}$ |
|---|---|---|---|---|---|---|
| | 288 | 13.86±0.85 | 13.83±3.23 | | 0.99 | 0.04 |
| | 293 | 12.55±0.64 | 14.29±2.18 | | 0.99 | 0.03 |
| | 298 | 9.82±0.70 | 11.50±1.04 | | 0.98 | 0.05 |
| R188A K299A | 303 | 4.86±0.44 | 8.56±0.20 | | 0.99 | 0.04 |
| | 288 | 2.79±0.20 | 7.93±0.47 | | 0.98 | 0.06 |
| | 293 | 3.26±0.09 | 7.39±0.50 | | 0.99 | 0.03 |
| | 298 | 6.53±0.2 | 8.97±0.15 | | 1 | 0 |
| R238A K243A | 303 | 7.63±0.26 | 10.78±2.19 | | 1 | 0.01 |
| | 288 | 20.53±2.25 | | | 1 | 0 |
| | 293 | 21.09±2.84 | | | 1 | 0 |
| | 298 | 19.80±1.71 | | | 1 | 0 |
| R188A R238A K243A | 303 | 13.12±0.81 | | | 1 | 0 |

These values represent the replicates with the poorest fits.

subunit interacts with one of the P-GlcN moieties of LPS whereas one of the K299 residues interacts with a carboxylic acid group of LPS molecule in the inner cavity. Therefore, introducing the R78A mutation will impact symmetrically equivalent binding sites. MsbA[R78A] showed a reduction in binding KDL[1-2] with ΔG ranging from –30.2±0.2 to -32.5±0.2 kJ/mol. At 298 K, KDL binding is enthalpically and entropically favorable whereas binding of the second KDL is similar to the wild-type protein (*Figure 2d*, *Figure 2—figure supplements 1–3*). The binding thermodynamics for MsbA[K299A] is reminiscent of the wild-type protein with a large, favorable change in entropy (-TΔS, –75±2 to -86±1 kJ/mol at 298 K) and unfavorable enthalpy (43±2–53±1 kJ/mol) (*Figure 2d*, *Figure 2—figure supplements 1–3*). The double mutant MsbA[R78A,K299A] protein shows a reduction in opposing entropic and enthalpic terms leading to an increase in ΔG by ~4 kJ/mol relative to wild-type MsbA (*Figure 2d*). Mutant cycle analysis indicates a coupling energy ($\Delta\Delta G_{int}$) of 1.7±0.4 kJ/mol that contributes to the stability of KDL-MsbA complex (*Figure 2e*, *Figure 2—figure supplements 1–3* and *Tables 2 and 3*). More generally, ΔΔ with a positive sign means favorable cooperation. Interestingly, the coupling enthalpy ($\Delta\Delta H_{int}$ of –26±15 kJ/mol) and coupling entropy ($\Delta(-T\Delta S)_{int}$ of 28±15 kJ/mol at 298 K) indicating that these residues contribute to KDL binding through an entropy driven process that overcomes an enthalpic barrier (*Figure 2e*, *Figure 2—figure supplements 1–3* and *Table 3*).

## KDL binding to the exterior binding site of MsbA

The recently discovered exterior KDL binding site (*Lyu et al., 2022*) located on the cytosolic leaflet of inner membrane has not been thoroughly investigated, prompting us to characterize this site by a triple mutant cycle (*Figure 3*, *Figure 3—figure supplements 1–6*). We first investigated R188 and K243, residues that both interact with one of the P-GlcN moieties of LOS. Like mutants targeting the interior LPS binding site, introducing mutants at the exterior site will impact binding at the two exterior sites. Both MsbA[R188A] and MsbA[K243A] single mutants marginally weakened the interaction by about 2 kJ/mol (*Figure 3a*, *Figure 3—figure supplements 1–6*). Enthalpy and entropy for KDL binding MsbA[R188A] and MsbA[K243A] was largely similar to the wild-type protein (*Figure 3a*, *Figure 3—figure supplements 1–6*). However, the R238A mutation significantly weakened the interaction with KDL, increasing ΔG by nearly 5 kJ/mol compared to the wild-type transporter (*Figure 3b*, *Figure 3—figure supplements 1–6* and *Table 3*) and resulted in an distinct thermodynamic pattern with negative enthalpy changes for both the first and second KDL binding events (*Figure 3a*, *Figure 3—figure supplements 1–6*). ΔG for MsbA[R188A,K243A] was comparable to the K243A single mutant form of the protein (*Figure 3a*, *Figure 3—figure supplements 1–6*). The positive coupling energy of 3.2±0.4 kJ/mol with contributions from a coupling enthalpy of 19±11 kJ/

**Table 2.** Thermodynamic signatures of KDL interacting with wild-type and mutant MsbA.

Reported are the mean with standard deviation (n=3), and the subscript denotes the $n^{th}$ KDL binding event.

| | T (K) | $\Delta G_1$ (kJ/mol) | $\Delta H_1$ (kJ/mol) | $-T\Delta S_1$ (kJ/mol) | $\Delta Cp_1$ (kJ/mol/K) | $\Delta G_2$ (kJ/mol) | $\Delta H_2$ (kJ/mol) | $-T\Delta S_2$ (kJ/mol) | $\Delta Cp_2$ (kJ/mol/K) | $\Delta G_3$ (kJ/mol) | $\Delta H_3$ (kJ/mol) | $-T\Delta S_3$ (kJ/mol) | $\Delta Cp_3$ (kJ/mol/K) |
|---|---|---|---|---|---|---|---|---|---|---|---|---|---|
| WT | 288 | −34.0±0.2 | −36.0±11.6 | 2.0±11.5 | 8.0±1.5 | −30.8±0.1 | −43.2±7.3 | 12.4±7.3 | 10.1±0.9 | −28.6±0.2 | −36.6±8.1 | 8.0±8.0 | 9.4±0.9 |
| | 293 | −34.4±0.1 | 4.3±5.4 | −38.6±5.4 | 7.4±1.5 | −31.1±0.2 | 7.9±2.7 | −39.0±2.7 | 9.1±0.8 | −29.0±0.2 | 10.9±3.3 | −39.8±3.3 | 8.5±0.6 |
| | 298 | −35.2±0.2 | 45.0±5.8 | −80.2±5.9 | 8.9±1.4 | −32.0±0.2 | 59.9±2.3 | −91.8±2.2 | 11.7±1.2 | −29.8±0.2 | 59.1±1.8 | −88.9±1.7 | 10.7±1.5 |
| | 303 | −37.0±0.3 | 86.1±12.1 | −123.0±12.4 | 8.3±1.5 | −34.1±0.2 | 112.3±7.1 | −146.4±7.1 | 10.6±1.0 | −31.8±0.2 | 107.7±7.1 | −139.5±7.1 | 9.8±1.1 |
| R78A | 288 | −31.6±0.2 | 9.6±4.5 | −41.2±4.7 | −2.6±1.1 | −28.8±0.3 | −13.0±19.1 | −15.8±18.8 | 5.0±1.7 | | | | |
| | 293 | −32.2±0.3 | −3.5±5.1 | −28.6±5.1 | −2.6±1.1 | −29.2±0.1 | 12.0±11.1 | −41.3±11.1 | 5.0±1.7 | | | | |
| | 298 | −32.5±0.2 | −16.7±9.6 | −15.9±9.6 | −2.6±1.1 | −30.2±0.2 | 37.1±5.9 | −67.2±6.1 | 5.0±1.7 | | | | |
| R188A | 288 | −31.2±0.2 | −23.0±8.2 | −8.3±8.1 | 6.4±1.1 | −27.4±0.4 | −39.1±0.5 | 11.8±0.7 | 11.2±1.0 | | | | |
| | 293 | −31.7±0.1 | 9.4±3.6 | −41.1±3.5 | 6.1±1.2 | −27.7±0.5 | 17.4±4.6 | −45.1±4.2 | 10.7±1.7 | | | | |
| | 298 | −32.6±0.1 | 42.1±3.7 | −74.6±3.7 | 6.9±0.9 | −28.8±0.2 | 74.4±8.8 | −103.2±8.6 | 12.1±0.7 | | | | |
| | 303 | −34.1±0.1 | 74.9±8.3 | −109.0±8.3 | 6.6±1.0 | −31.1±0.1 | 131.6±12.8 | −162.7±12.8 | 11.5±0.7 | | | | |
| R238A | 288 | −30.9±0.1 | −66.8±3.4 | 35.9±3.3 | 4.8±0.3 | −29.4±0.2 | −57.9±17.9 | 28.6±17.8 | 2.8±2.4 | | | | |
| | 293 | −30.6±0.1 | −42.6±4.0 | 12.1±3.9 | 4.1±0.3 | −29.2±0.2 | −43.2±6.5 | 13.9±6.5 | 0.9±2.5 | | | | |
| | 298 | −30.4±0.1 | −17.9±4.8 | −12.6±4.8 | 5.8±0.3 | −28.6±0.2 | −26.8±7.2 | −1.9±7.0 | 5.5±2.4 | | | | |
| | 303 | −30.9±0.1 | 7.2±5.8 | −38.1±5.9 | 5.1±0.3 | −29.0±0.1 | −9.5±18.6 | −19.5±18.7 | 3.7±2.4 | | | | |
| K243A | 288 | −32.6±0.1 | −48.5±7.3 | 15.9±7.2 | 9.9±0.9 | −28.4±0.3 | −31.6±7.3 | 3.3±7.0 | 8.5±0.9 | | | | |
| | 293 | −32.8±0.2 | 1.5±3.5 | −34.4±3.5 | 9.1±0.8 | −28.8±0.2 | 11.7±3.1 | −40.5±3.0 | 7.3±1.2 | | | | |
| | 298 | −33.6±0.1 | 52.2±3.9 | −85.9±4.0 | 11.2±1.2 | −29.6±0.2 | 56.0±2.0 | −85.5±2.1 | 10.3±0.6 | | | | |
| | 303 | −35.6±0.2 | 103.3±8.0 | −138.9±8.2 | 10.3±1.0 | −31.5±0.2 | 100.8±5.6 | −132.3±5.8 | 9.1±0.8 | | | | |

*Table 2 continued on next page*

eLife Research article

Biochemistry and Chemical Biology | Structural Biology and Molecular Biophysics

Table 2 continued

| | T (K) | $\Delta G_1$ (kJ/mol) | $\Delta H_1$ (kJ/mol) | $-T\Delta S_1$ (kJ/mol) | $\Delta Cp_1$ (kJ/mol/K) | $\Delta G_2$ (kJ/mol) | $\Delta H_2$ (kJ/mol) | $-T\Delta S_2$ (kJ/mol) | $\Delta Cp_2$ (kJ/mol/K) | $\Delta G_3$ (kJ/mol) | $\Delta H_3$ (kJ/mol) | $-T\Delta S_3$ (kJ/mol) | $\Delta Cp_3$ (kJ/mol/K) |
|---|---|---|---|---|---|---|---|---|---|---|---|---|---|
| K299A | 288 | −31.1±0.1 | −22.2±9.5 | −8.9±9.4 | 6.4±1.1 | −28.8±0.1 | −18.8±9.6 | −10.0±9.5 | 7.3±1.0 | −26.4±0.9 | −35.6±71.5 | 9.3±70.7 | 11.4±7.1 |
| | 293 | −31.6±0.1 | 9.9±3.8 | −41.5±3.9 | 5.6±1.1 | −29.4±0.1 | 18.0±4.6 | −47.4±4.6 | 6.0±1.0 | −27.0±0.5 | 22.7±37.2 | −49.6±37.5 | 9.2±8.6 |
| | 298 | −32.4±0.1 | 42.7±2.1 | −75.1±1.9 | 7.4±1.2 | −30.2±0.1 | 55.9±1.7 | −86.1±1.6 | 9.0±1.1 | −27.7±0.3 | 82.9±7.8 | −110.6±7.9 | 14.7±5.1 |
| | 303 | −34.0±0.1 | 75.8±7.7 | −109.8±7.7 | 6.7±1.2 | −32.1±0.1 | 94.3±6.3 | −126.4±6.3 | 7.8±1.1 | −30.4±0.2 | 144.1±30.5 | −174.5±30.3 | 12.5±6.4 |
| R78A K299A | 288 | −30.9±0.2 | −36.8±5.6 | 5.9±5.4 | 4.4±0.8 | −28.2±0.1 | −49.1±5.7 | 20.9±5.7 | 12.0±1.4 | | | | |
| | 293 | −30.9±0.1 | −15.0±2.8 | −16.0±2.8 | 4.4±0.8 | −28.3±0.1 | 10.7±9.6 | −39.1±9.7 | 12.0±1.4 | | | | |
| | 298 | −31.4±0.1 | 6.8±3.8 | −38.2±3.9 | 4.4±0.8 | −29.5±0.3 | 70.6±15.6 | −100.1±15.9 | 12.0±1.4 | | | | |
| R188A R238A | 288 | −29.0±0.4 | −93.3±15.2 | 64.3±14.8 | 7.8±1.3 | | | | | | | | |
| | 293 | −28.5±0.2 | −53.3±9.2 | 24.8±9.1 | 5.9±1.6 | | | | | | | | |
| | 298 | −27.9±0.1 | −11.6±4.7 | −16.3±4.6 | 10.6±1.0 | | | | | | | | |
| | 303 | −28.8±0.1 | 30.9±5.9 | −59.7±5.9 | 8.7±1.2 | | | | | | | | |
| R188A K243A | 288 | −33±0.1 | −20.3±6.0 | −12.6±6.0 | 4.9±0.6 | −28.9±0.1 | −21.3±3.4 | −7.6±3.3 | 4.8±0.5 | | | | |
| | 293 | −33.5±0.1 | 4.5±3.2 | −38.0±3.2 | 4.0±0.6 | −29.3±0.1 | 2.7±3.4 | −32.0±3.4 | 4.2±0.4 | | | | |
| | 298 | −34.1±0.1 | 30.1±1.9 | −64.3±1.9 | 6.2±0.7 | −29.9±0.1 | 27.3±5.0 | −57.1±5.0 | 5.6±0.6 | | | | |
| | 303 | −35.5±0.1 | 56.1±4.1 | −91.7±4.1 | 5.3±0.6 | −31.1±0.1 | 52.0±7.3 | −83.2±7.4 | 5.0±0.5 | | | | |
| R188A K299A | 288 | −26.8±0.1 | −15.4±16.2 | −11.4±16.0 | 8.9±2.2 | −26.9±0.6 | −15.7±36.0 | −11.1±35.4 | 5.2±3.8 | | | | |
| | 293 | −27.5±0.1 | 29.5±5.2 | −57.0±5.2 | 7.8±1.3 | −27.2±0.4 | 10.1±18.9 | −37.3±18.9 | 5.8±6.0 | | | | |
| | 298 | −28.6±0.2 | 75.2±7.6 | −103.8±7.4 | 10.4±3.5 | −28.2±0.2 | 35.4±6.4 | −63.6±6.3 | 4.4±1.4 | | | | |
| | 303 | −30.8±0.2 | 121.4±19.9 | −152.2±20.0 | 9.4±2.6 | −29.4±0.1 | 60.5±14.2 | −89.9±14.2 | 5.0±2.8 | | | | |

*Table 2 continued*

| | T (K) | $\Delta G_1$ (kJ/mol) | $\Delta H_1$ (kJ/mol) | $-T\Delta S_1$ (kJ/mol) | $\Delta Cp_1$ (kJ/mol/K) | $\Delta G_2$ (kJ/mol) | $\Delta H_2$ (kJ/mol) | $-T\Delta S_2$ (kJ/mol) | $\Delta Cp_2$ (kJ/mol/K) | $\Delta G_3$ (kJ/mol) | $\Delta H_3$ (kJ/mol) | $-T\Delta S_3$ (kJ/mol) | $\Delta Cp_3$ (kJ/mol/K) |
|---|---|---|---|---|---|---|---|---|---|---|---|---|---|
| | 288 | −30.7±0.2 | −42.4±8.3 | 11.8±8.1 | −1.2±0.5 | −28.1±0.1 | 13.2±9.6 | −41.4±9.6 | −3.8±2.4 | | | | |
| | 293 | −30.8±0.1 | −46.9±6.0 | 16.1±6.0 | −4.7±0.3 | −28.8±0.2 | −5.4±4.5 | −23.4±4.5 | −4.7±1.6 | | | | |
| | 298 | −29.6±0.1 | −48.4±4.0 | 18.8±4.1 | 3.8±0.8 | −28.8±0.1 | −23.3±17.2 | −5.5±17.2 | −2.5±3.8 | | | | |
| R238A K243A | 303 | −29.7±0.1 | −48.3±3.4 | 18.6±3.4 | 0.4±0.6 | −28.9±0.6 | −40.7±31 | 11.9±31.5 | −3.4±2.9 | | | | |
| | 288 | −25.9±0.3 | −25.2±5.7 | −0.7±5.4 | 6.2±0.3 | | | | | | | | |
| | 293 | −26.3±0.3 | 6.5±3.7 | −32.7±3.4 | 5.3±1.4 | | | | | | | | |
| | 298 | −26.9±0.2 | 38.9±0.6 | −65.7±0.7 | 7.6±2.5 | | | | | | | | |
| R188A R238A K243A | 303 | −28.3±0.2 | 71.7±3.3 | −100±3.3 | 6.7±0.9 | | | | | | | | |

**Table 3.** Double mutant cycle analysis of the first KDL binding to wild-type and mutant MsbA.
The ΔΔ values mutant relative to the wild-type protein. Reported are the mean (n=3).

| | | Temperature (K) | ΔΔG (kJ/mol) | ΔΔH (kJ/mol) | Δ(-TΔS) (kJ/mol) | ΔΔG$_{int}$ (kJ/mol) | ΔΔH$_{int}$ (kJ/mol) | Δ(-ΔTS)$_{int}$ (kJ/mol) |
|---|---|---|---|---|---|---|---|---|
| | | 288 | 2.4±0.2 | 45.6±12.5 | −43.2±12.4 | | | |
| | | 293 | 2.2±0.2 | −7.8±7.5 | 10.0±7.5 | | | |
| R78A | | 298 | 2.7±0.2 | −61.7±11.3 | 64.4±11.3 | | | |
| | | 288 | 2.8±0.2 | 13.0±14.2 | −10.2±14.0 | | | |
| | | 293 | 2.7±0.1 | 5.2±6.5 | −2.5±6.5 | | | |
| | | 298 | 2.7±0.2 | −3.0±6.9 | 5.6±7.0 | | | |
| R188A | | 303 | 2.9±0.4 | −11.2±14.7 | 14.1±14.9 | | | |
| | | 288 | 3.1±0.2 | −30.9±12.1 | 33.9±11.9 | | | |
| | | 293 | 3.8±0.1 | −46.9±6.7 | 50.7±6.7 | | | |
| | | 298 | 4.8±0.1 | −62.9±7.5 | 67.7±7.6 | | | |
| R238A | | 303 | 6.1±0.4 | −78.8±13.5 | 84.9±13.7 | | | |
| | | 288 | 1.4±0.2 | −12.5±13.7 | 13.9±13.5 | | | |
| | | 293 | 1.5±0.2 | −2.7±6.5 | 4.3±6.5 | | | |
| | | 298 | 1.6±0.2 | 7.2±7.0 | −5.6±7.1 | | | |
| K243A | | 303 | 1.4±0.4 | 17.3±14.5 | −15.9±14.8 | | | |
| | | 288 | 2.9±0.2 | 13.8±15.1 | −10.9±14.8 | | | |
| | | 293 | 2.8±0.1 | 5.7±6.6 | −2.9±6.7 | | | |
| | | 298 | 2.8±0.2 | −2.4±6.1 | 5.2±6.2 | | | |
| K299A | | 303 | 3.0±0.4 | −10.3±14.3 | 13.3±14.6 | | | |
| | | 288 | 3.2±0.2 | −0.8±12.9 | 4.0±12.6 | 2.2±0.5 | 60.2±23.4 | −58.0±23.1 |
| | | 293 | 3.4±0.1 | −19.2±6.1 | 22.7±6.1 | 1.6±0.4 | 17.1±11.6 | −15.6±11.8 |
| R78A | K299A | 298 | 3.8±0.2 | −38.2±6.9 | 42.0±7.1 | 1.7±0.4 | −25.8±14.6 | 27.5±14.7 |
| | | 288 | 5.0±0.5 | −57.4±19.1 | 62.4±18.7 | 0.9±0.6 | 39.5±26.7 | −38.7±26.2 |
| | | 293 | 5.9±0.2 | −57.5±10.7 | 63.5±10.5 | 0.6±0.4 | 15.8±14.2 | −15.2±14.1 |
| | | 298 | 7.3±0.2 | −56.6±7.5 | 64.0±7.5 | 0.1±0.4 | −9.2±12.6 | 9.3±12.7 |
| R188A | R238A | 303 | 8.2±0.4 | −55.1±13.5 | 63.4±13.7 | 0.7±0.5 | −34.9±24.0 | 35.6±24.5 |
| | | 288 | 1.0±0.2 | 15.7±13.1 | −14.6±12.9 | 3.2±0.5 | −15.1±23.8 | 18.3±23.4 |
| | | 293 | 0.9±0.1 | 0.3±6.2 | 0.6±6.2 | 3.4±0.2 | 2.2±11.1 | 1.2±11.1 |
| | | 298 | 1.1±0.2 | −14.9±6.0 | 16.0±6.2 | 3.2±0.4 | 19.1±11.5 | −16.0±11.8 |
| R188A | K243A | 303 | 1.4±0.4 | −29.9±12.7 | 31.4±13.0 | 2.8±0.6 | 36.0±24.2 | −33.2±24.7 |
| | | 288 | 7.2±0.2 | 20.6±20.0 | −13.4±19.7 | −1.5±0.5 | 6.2±28.8 | −7.7±28.4 |
| | | 293 | 6.9±0.1 | 25.2±7.5 | −18.3±7.5 | −1.4±0.2 | −14.4±11.9 | 13.0±11.9 |
| | | 298 | 6.6±0.2 | 30.2±9.6 | −23.6±9.6 | −1.1±0.4 | −35.5±13.2 | 34.3±13.3 |
| R188A | K299A | 303 | 6.1±0.4 | 35.3±23.3 | −29.2±23.5 | −0.3±0.6 | −56.8±31.0 | 56.5±31.5 |
| | | 288 | 3.4±0.2 | −6.4±14.3 | 9.8±14.1 | 1.1±0.5 | −36.9±23.3 | 38.0±22.8 |
| | | 293 | 3.6±0.1 | −51.1±8.1 | 54.7±8.1 | 1.8±0.2 | 1.5±12.4 | 0.3±12.4 |
| | | 298 | 5.6±0.2 | −93.4±7.0 | 99.0±7.2 | 0.8±0.4 | 37.7±12.4 | −37.0±12.7 |
| R238A | K243A | 303 | 7.3±0.4 | −134.4±12.5 | 141.6±12.9 | 0.2±0.6 | 72.8±23.4 | −72.6±24.0 |

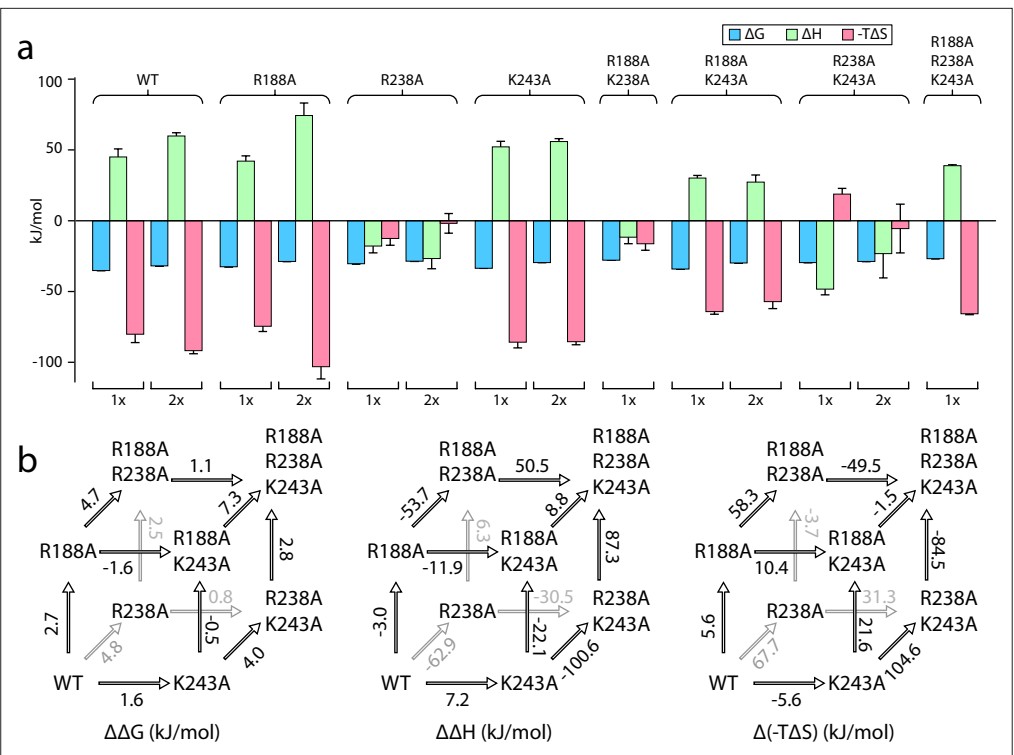

**Figure 3.** Triple mutant cycle analysis of the exterior LPS binding site of MsbA. (**a**) Thermodynamics for MsbA and mutants (MsbA[R188A], MsbA[R238A], MsbA[K243A], MsbA[R188A,R243A], MsbA[R188A,K243A], MsbA[R238A,R243A], and MsbA[R188A,R238A,K299A]) binding KDL at 298 K. (**b**) Triple mutant cycles for MsbA and mutants with (from left to right) ΔΔG, ΔΔH and Δ(-TΔS) values indicated over the respective arrows. Shown are values at 298 K. Reported are the average and standard deviation from repeated measurements (n = 3).

The online version of this article includes the following figure supplement(s) for figure 3:

**Figure supplement 1.** Representative native mass spectra MsbA mutants in the presence of 0.8 µM KDL.

**Figure supplement 2.** Representative native mass spectra MsbA double mutants in the presence of 0.8 µM KDL.

**Figure supplement 3.** Determination of equilibrium dissociation constants (K_D) for MsbA mutants binding KDL.

**Figure supplement 4.** Determination of equilibrium dissociation constants (K_D) for MsbA mutants binding KDL.

**Figure supplement 5.** Determination of thermodynamic parameters for KDL binding wild-type and mutant MsbA.

**Figure supplement 6.** Determination of thermodynamic parameters for KDL binding wild-type and mutant MsbA.

mol and a coupling entropy of –16±12 kJ/mol at 298 K (*Figure 3b*, *Figure 3—figure supplements 1–6* and *Table 3*). Combining mutation R238A with R188A, MsbA[R188A,R238A] decreased ΔH by 57 kJ/mol at the cost of increasing -TΔS by 64 kJ/mol at 298 K (*Figure 3b*, *Figure 3—figure supplements 1–6* and *Table 3*). The coupling energy for R188A and R238A is approximately zero as a result of equal coupling enthalpy and entropy of different signs. Compared to the wild-type protein, MsbA[R238A,K243A] results in an inversion of the thermodynamic signature with binding now being driven by enthalpy. More specifically, this inversion is accompanied by ΔΔH and Δ(-TΔS) of –93±7 kJ/mol and 99±7 kJ/mol at 298 K (*Figure 3b*, *Figure 3—figure supplements 1–6* and *Table 3*). Again, the coupling enthalpy and entropy (at 298 K) of equal magnitude but opposite signs give rise to a coupling energy of zero for R238A and R243A (*Figure 3b*, *Figure 3—figure supplements 1–6* and *Table 3*). Introduction of the R188A mutation into MsbA[R238A,K243A], results in reversal of the thermodynamic signature to mirror that of MsbA[R188A,K243A] (*Figure 3a*, *Figure 3—figure supplements 1–6*). The coupling energy, coupling enthalpy, and coupling entropy for R188A, R238A, and R243A are 3.4±0.5 kJ/mol, 100±16 kJ/mol, and –97±16 kJ/mol at 298 K (*Table 4*), respectively. Taken together, these results demonstrate KDL binding to MsbA is sensitive to mutations at both the interior and exterior sites.

**Table 4.** Triple mutant cycle analysis of the first KDL binding to wild-type and mutant MsbA. Shown as described in *Table 3*.

| | | | Temperature (K) | ΔΔG (kJ/mol) | ΔΔH (kJ/mol) | Δ(-TΔS) (kJ/mol) | ΔΔG$_{int}$ (kJ/mol) | ΔΔH$_{int}$ (kJ/mol) | Δ(-ΔTS)$_{int}$ (kJ/mol) |
|---|---|---|---|---|---|---|---|---|---|
| | R238A | | 288 | 2.2±0.2 | −70.4±8.9 | 72.6±8.7 | | | |
| | | | 293 | 3.2±0.1 | −62.7±5.4 | 65.9±5.3 | | | |
| | | | 298 | 4.7±0.1 | −53.7±6.1 | 58.3±6.1 | | | |
| | | | 303 | 5.4±0.1 | −43.9±10.2 | 49.3±10.3 | | | |
| | K243A | | 288 | −1.8±0.2 | 2.6±11.0 | 19.9±10.8 | | | |
| | | | 293 | −1.8±0.2 | −4.9±5.0 | −6.4±5.0 | | | |
| | | | 298 | −1.6±0.1 | −11.9±5.4 | 10.4±5.4 | | | |
| | | | 303 | −1.4±0.2 | −18.7±11.5 | 17.3±11.8 | | | |
| R188A | R238A | K243A | 288 | 5.3±0.4 | −2.2±10.0 | 7.5±9.7 | −4.9±0.5 | −65.6±17.4 | 85.0±16.9 |
| | | | 293 | 5.4±0.4 | −2.9±5.1 | 8.4±4.9 | −4.0±0.5 | −64.7±8.9 | 51.2±8.8 |
| | | | 298 | 5.7±0.2 | −3.2±3.8 | 8.9±3.7 | −2.6±0.2 | −62.4±8.9 | 59.8±8.9 |
| | | | 303 | 5.8±0.2 | −3.2±8.9 | 9.0±8.9 | −1.8±0.4 | −59.5±17.8 | 57.7±18.0 |
| | R188A | | 288 | 1.9±0.2 | −26.5±8.9 | 28.4±8.7 | | | |
| | | | 293 | 2.1±0.1 | −10.6±5.4 | 12.8±5.3 | | | |
| | | | 298 | 2.5±0.1 | 6.3±6.1 | −3.7±6.1 | | | |
| | | | 303 | 2.2±0.1 | 23.7±10.2 | −21.5±10.3 | | | |
| | K243A | | 288 | 0.3±0.2 | 24.4±8.0 | −24.1±7.8 | | | |
| | | | 293 | −0.2±0.1 | −4.2±5.3 | 4.0±5.3 | | | |
| | | | 298 | 0.8±0.1 | −30.5±6.2 | 31.3±6.2 | | | |
| | | | 303 | 1.2±0.2 | −55.5±9.9 | 56.7±10.2 | | | |
| R238A | R188A | K243A | 288 | 5.1±0.2 | 41.7±6.6 | −36.6±6.4 | −2.9±0.4 | −43.8±13.7 | 40.9±13.3 |
| | | | 293 | 4.3±0.4 | 49.1±5.4 | −44.8±5.1 | −2.4±0.4 | −64.0±9.3 | 61.6±9.1 |
| | | | 298 | 3.6±0.2 | 56.7±4.9 | −53.2±4.9 | −0.2±0.2 | −81.0±10.0 | 80.8±10.0 |
| | | | 303 | 2.6±0.2 | 64.5±6.7 | −61.9±6.9 | 0.8±0.4 | −96.3±15.7 | 97.1±16.0 |
| | R188A | | 288 | −0.4±0.2 | 28.1±11.0 | −28.5±10.8 | | | |
| | | | 293 | −0.7±0.2 | 3.0±5.0 | −3.7±5.0 | | | |
| | | | 298 | −0.5±0.1 | −22.1±5.4 | 21.6±5.4 | | | |
| | | | 303 | 0.1±0.2 | −47.2±11.5 | 47.3±11.8 | | | |
| | R238A | | 288 | 1.9±0.2 | 6.1±8.0 | −4.1±7.8 | | | |
| | | | 293 | 2.0±0.1 | −48.4±5.3 | 50.4±5.3 | | | |
| | | | 298 | 4.0±0.1 | −100.6±6.2 | 104.6±6.2 | | | |
| | | | 303 | 5.9±0.2 | −151.6±9.9 | 157.5±10.2 | | | |
| K243A | R188A | R238A | 288 | 6.7±0.2 | 23.3±9.2 | −16.6±8.9 | −5.2±0.4 | 10.9±16.4 | −16.0±16.0 |
| | | | 293 | 6.6±0.4 | 5.0±5.1 | 1.6±4.9 | −5.2±0.5 | −50.4±8.9 | 45.1±8.8 |
| | | | 298 | 6.8±0.2 | −13.3±3.9 | 20.1±4.0 | −3.3±0.2 | −109.4±9.2 | 106.1±9.2 |
| | | | 303 | 7.3±0.2 | −31.6±8.7 | 38.9±8.9 | −1.3±0.4 | −167.2±17.5 | 165.9±17.9 |

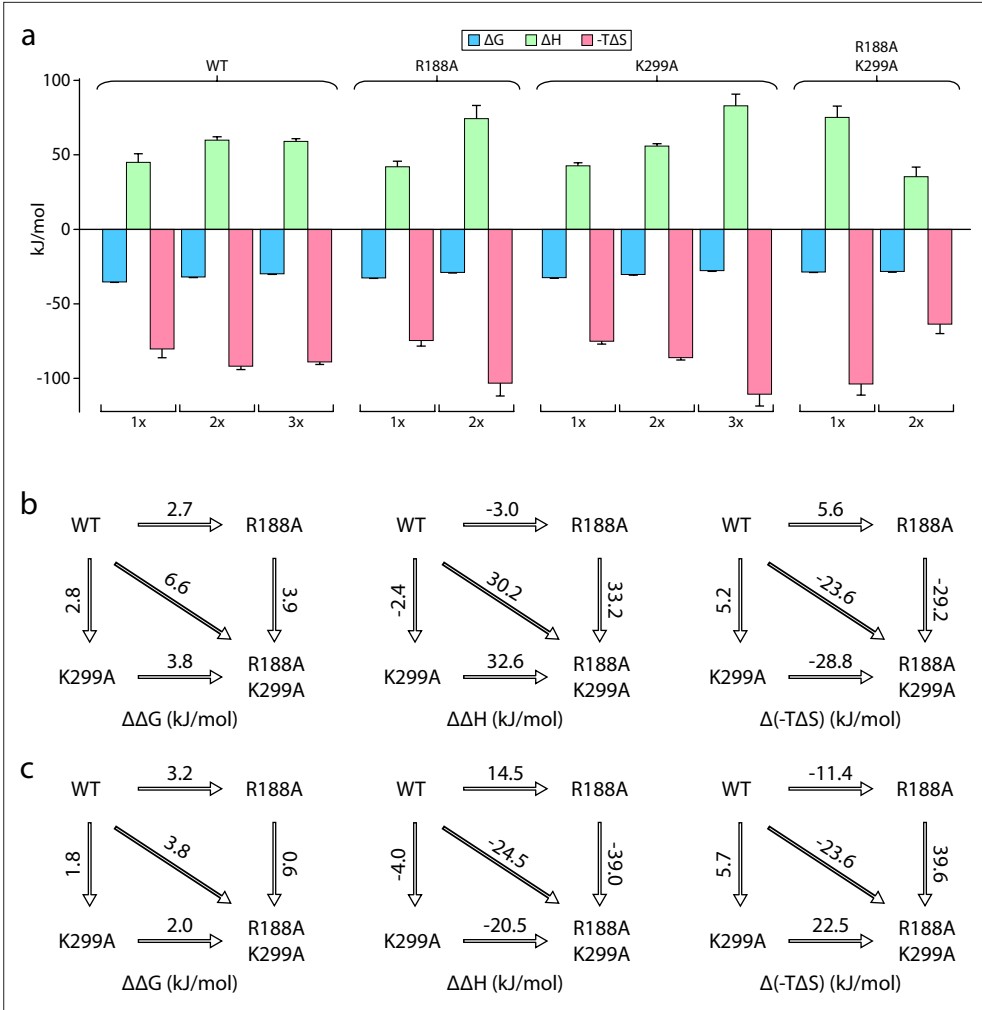

**Figure 4.** Mutant cycle of MsbA residues located within the interior and exterior LOS binding sites.
(**a**) Thermodynamic signatures for MsbA and mutants binding KDL at 298 K. (**b–c**) Double mutant cycle analysis for R188 and K299. Shown are results for the first (panel b) and second (panel c) KDL binding to MsbA. Shown from left to right is ΔΔG, ΔΔH and Δ(-TΔS) and the values indicated over the respective arrows at 298 K. Reported are the average and standard deviation from repeated measurements (n = 3).

## Dissecting KDL binding to the interior and exterior site(s) of MsbA

An open question is if the interior and exterior LOS binding sites of MsbA are allosterically coupled? We focused on the R188A and K299A mutants located at the exterior and interior binding sites, respectively. Results for both single mutants were presented above. MsbA containing the R188A and K299A mutations drastically reduced the binding of KDL (*Figure 4a*). The ΔG for MsbA[R188A,K299A] increased by more than 6 kJ/mol compared to the wild-type protein (*Figure 4b*). This approximately doubles compared to MsbA containing either of the single point mutations. Mutant cycle analysis revealed a negative coupling energy of −1.1±0.4 kJ/mol that partitioned into a coupling enthalpy of −36±13 kJ/mol and coupling entropy of 34±13 kJ/mol at 298 K (*Figure 4b* and *Table 3*). In short, mutations at either LOS binding site have a negative impact on binding that is accompanied by a gain in both favorable entropy and unfavorable enthalpy.

## Mutant cycle analysis of KDL binding to vanadate-trapped MsbA

As MsbA, like other ABC transporters, is highly dynamic, we sought to trap the transporter in an OF conformation using ADP and vanadate to interrogate binding at the exterior lipid binding site. We characterized the binding of KDL to vanadate-trapped MsbA and proteins containing single R188A, R238A, and K243A mutations (*Table 5*). Here, we focused on the binding of the first and second lipid,

**Table 5.** Equilibrium dissociation constants ($K_D$) for KDL binding MsbA trapped with ADP and vanadate at various temperatures. Reported are the mean and standard deviation (n=3).

| | Temperature(K) | $K_{D1}$(µM) | $K_{D2}$(µM) | $K_{D3}$(µM) | $R^{2*}$ | $X^{2*}$ |
|---|---|---|---|---|---|---|
| | 293 | 0.51±0.04 | 1.16±0.07 | | 0.97 | 0.08 |
| | 298 | 0.44±0.02 | 0.93±0.09 | | 0.94 | 0.17 |
| | 303 | 0.38±0.02 | 0.73±0.05 | | 0.94 | 0.15 |
| WT | 310 | 0.31±0.01 | 0.53±0.04 | | 0.92 | 0.2 |
| | 293 | 1.56±0.09 | 3.46±0.15 | 10.98±1.15 | 0.98 | 0.07 |
| | 298 | 1.53±0.11 | 3.40±0.21 | 10.15±1.62 | 0.98 | 0.06 |
| | 303 | 1.35±0.10 | 2.90±0.26 | 9.02±1.11 | 0.98 | 0.05 |
| R188A | 310 | 0.93±0.09 | 1.89±0.24 | 5.50±1.10 | 0.97 | 0.07 |
| | 293 | 1.67±0.23 | 6.71±1.38 | | 0.99 | 0.05 |
| | 298 | 0.91±0.10 | 3.15±0.12 | | 0.98 | 0.06 |
| | 303 | 0.34±0.04 | 1.09±0.23 | 3.75±0.60 | 0.95 | 0.11 |
| R238A | 310 | 0.10±0.02 | 0.33±0.11 | 0.92±0.38 | 1 | 0 |
| | 293 | 2.01±0.06 | 5.26±0.85 | | 0.99 | 0.02 |
| | 298 | 1.46±0.01 | 3.90±0.48 | | 0.99 | 0.03 |
| | 303 | 0.60±0.02 | 1.54±0.12 | 3.72±0.54 | 0.98 | 0.04 |
| K243A | 310 | 0.25±0.02 | 0.46±0.02 | 1.06±0.04 | 0.89 | 0.2 |
| | 293 | 1.24±0.10 | 4.82±0.16 | | 0.99 | 0.03 |
| | 298 | 1.17±0.11 | 4.65±0.92 | | 0.99 | 0.02 |
| | 303 | 0.87±0.06 | 3.05±0.26 | | 0.98 | 0.04 |
| R188A R238A | 310 | 0.39±0.05 | 1.28±0.09 | 6.36±1.07 | 0.98 | 0.04 |
| | 293 | 3.64±0.28 | 15.66±4.76 | | 0.99 | 0.03 |
| | 298 | 2.23±0.07 | 6.68±0.17 | | 0.99 | 0.03 |
| | 303 | 1.06±0.05 | 3.10±0.33 | 19.33±3.70 | 0.99 | 0.03 |
| R188A K243A | 310 | 0.66±0.02 | 1.56±0.04 | 5.70±0.93 | 0.98 | 0.04 |
| | 293 | 1.31±0.04 | 4.35±0.07 | | 0.99 | 0.03 |
| | 298 | 1.03±0.09 | 3.65±0.15 | | 0.99 | 0.02 |
| | 303 | 0.48±0.05 | 1.97±0.09 | | 0.97 | 0.09 |
| R238A K243A | 310 | 0.10±0.02 | 0.58±0.06 | | 0.74 | 0.93 |

These values represent the replicates with the poorest fits.

since MsbA has two, symmetrically related KDL binding sites in the open, OF conformation. Thermodynamics of MsbA(KDL)$_{1-2}$ binding is like the non-trapped transporter, wherein entropy (-T∆S ranging from –58±1 to -69±1 kJ/mol at 298 K) is more favorable than a positive enthalpic term (∆H ranging from 22±1–35±1 kJ/mol) (*Figure 5a*, *Figure 5—figure supplements 1–6* and *Table 6*). The single mutant proteins (MsbA[R188A], MsbA[R238A], and MsbA[K243A]) showed a slight increase in ∆G (at most 5 kJ/mol) (*Figure 5a*, *Figure 5—figure supplements 1–6*). Notably, we found MsbA[R238A] and MsbA[K243A] had about a four-fold increase in ∆H and favorable entropy was about two-fold higher (*Figure 5a*, *Figure 5—figure supplements 1–6*). Double mutant cycle analysis of the pairwise mutants revealed a positive coupling energy of ~2 kJ/mol for MsbA binding one and two KDLs (*Figure 5b*, *Figure 5—figure supplements 1–6* and *Table 7*). Focusing on the first KDL binding event, the coupling enthalpy and coupling entropy at 298 K for R188 and K238 was 89±7 kJ/mol and –87±7 kJ/mol, respectively

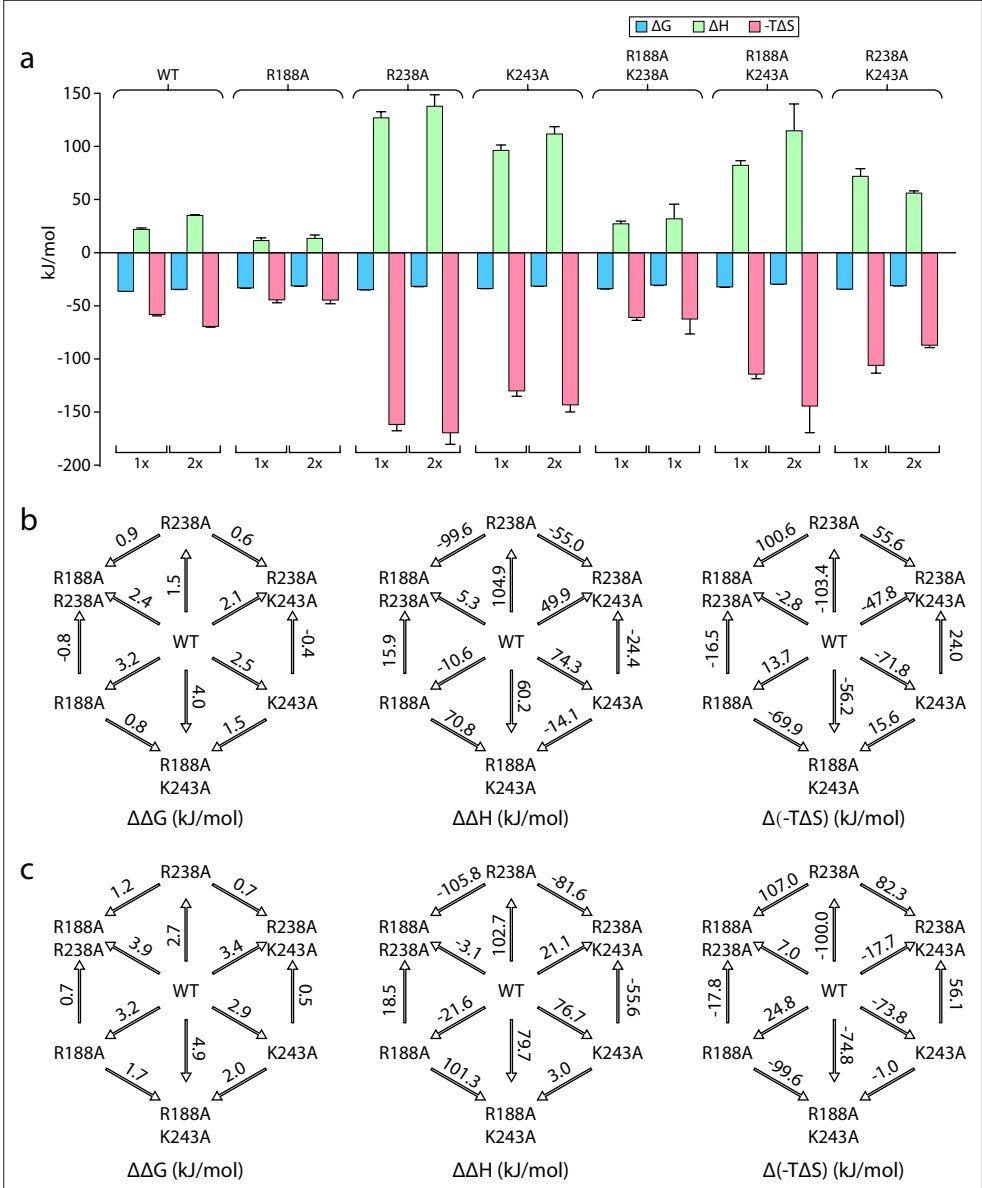

**Figure 5.** Double mutant cycles reveal thermodynamic insight into KDL binding vanadate-trapped MsbA. (**a**) Thermodynamic signatures for MsbA and mutants binding KDL at 298 K. (**b–c**) Double mutant cycle analysis for pairs of R188, R238, and K243 with a total of three combinations. Shown are results for the first (panel b) and second (panel c) KDL binding to MsbA trapped in an open, OF conformation with ADP and vanadate. Within each panel, ΔΔG, ΔΔH and Δ(-TΔS) are shown from left to right and their values at 298 K are indicated over the respective arrows. Reported are the average and standard deviation from repeated measurements (n = 3).

The online version of this article includes the following figure supplement(s) for figure 5:

**Figure supplement 1.** Representative native mass spectra wild-type and mutant MsbA trapped with vanadate and ADP in the presence of 0.8 μM KDL.

**Figure supplement 2.** Representative native mass spectra MsbA double mutants trapped with vanadate and ADP in the presence of 0.8 μM KDL.

**Figure supplement 3.** Determination of equilibrium dissociation constants ($K_D$) for KDL binding wild-type and mutant MsbA trapped with ADP and vanadate.

**Figure supplement 4.** Determination of equilibrium dissociation constants ($K_D$) for KDL binding MsbA double mutants trapped with ADP and vanadate.

*Figure 5 continued on next page*

*Figure 5 continued*

**Figure supplement 5.** Determination of thermodynamic parameters for KDL binding wild-type and mutant MsbA trapped with ADP and vanadate.

**Figure supplement 6.** Determination of thermodynamic parameters for KDL binding MsbA double mutants trapped with ADP and vanadate.

(*Figure 5b*, *Figure 5—figure supplements 1–6* and *Tables 7 and 8*). Likewise, R238 and K243 showed 129±11 kJ/mol of coupling enthalpy and –127±11 kJ/mol of coupling entropy at 298 K (*Figure 5b*, *Figure 5—figure supplements 1–6* and *Tables 7 and 8*). However, the R188 and K243 pair revealed a relatively low coupling enthalpy and coupling entropy at 298 K of 3.5±7 kJ/mol and 2±7 kJ/mol, respectively (*Figure 5b*, *Figure 5—figure supplements 1–6* and *Tables 7 and 8*). These results highlight the importance of entropic and enthalpic contributions that underpin specific lipid binding sites.

## Discussion

Thermodynamics provide unique insight into the molecular forces that drive specific MsbA-KDL interactions. A recurring thermodynamic strategy for specific KDL-MsbA interactions is a large, favorable entropic term that opposes a positive enthalpic value. The human G-protein-gated inward rectifier potassium channel (Kir3.2) also used a similar thermodynamic strategy to engage phosphoinositides (PIPs) (*Qiao et al., 2021*). The large, positive entropy could stem from solvent reorganization of the lipid with a carbohydrate containing headgroup, and desolvation of hydrated binding pockets on the membrane protein. The release of ordered solvent to the bulk solvent would contribute favorably to entropy. These experiments are performed in detergent and reorganization of detergent may also play a role. Previous work has shown soluble protein-ligand interactions can be driven by a large, positive entropy term that outweighs a large, positive enthalpic penalty (*Frederick et al., 2007*; *Tzeng and Kalodimos, 2009*; *Tzeng and Kalodimos, 2012*; *Caro et al., 2017*). In these cases, the reaction is mainly driven by conformational entropy originating in enhanced protein motions. However, it is unclear if the conformational dynamics of MsbA are enhanced when bound to KDL.

Most of the van' t Hoff plots followed non-linear trends, indicating $C_p$ is not constant over the selected temperature range (*Prabhu and Sharp, 2005*). In nearly all cases, a positive $\Delta C_p$ was observed that ranged in value from 4 to 12 kJ/mol·K (*Tables 2 and 6*). Solvation of polar groups in aqueous solvent has been ascribed to positive heat capacities whereas the collapse of apolar residues from their solvated state is accompanied by a negative change in heat capacity (*Prabhu and Sharp, 2005*; *Makhatadze and Privalov, 1995*). Reorganization of the hydrated, polar headgroups of KD is consistent with the positive heat capacity observed here. However, change in heat capacity could also be ascribed to temperature-dependent conformational changes in MsbA and/or KDL. Notably, vanadate-trapped MsbA locked in an open, OF conformation should be less conformationally dynamic than the apo protein, which is known to adopt a number of open, IF conformations where the NBDs are separated by different distances. Similar positive heat capacities were observed for the different conformations, suggesting the dynamics of MsbA marginally contribute to the observed non-linear trends. Notably, the headgroup of KDL is nestled in a hydrophilic, basic patch of MsbA in the open, OF conformation. Similarly, the headgroup of PIP binds a hydrophilic, basic pocket in Kir3.2. These hydrophilic patches will be highly solvated, which will be desolvated upon binding lipids contributing favorably to entropy.

Thermodynamics of MsbA-lipid interactions contrast those observed for a different membrane protein. Phospholipid binding to the bacterial ammonia channel (AmtB) were largely driven by enthalpy and, in most cases, entropy was unfavorable (*Cong et al., 2016*). Another interesting observation for AmtB-lipid interactions was significant enthalpy-entropy compensation for each sequential lipid binding event. Here, enthalpy-entropy compensation is not as pronounced. This result may reflect the much higher affinity and specific MsbA-KDL interactions compared to the weaker AmtB-lipid interactions, sometimes referred to as non-annular lipids (*Bolla et al., 2019*). Moreover, we have focused the titration here to characterize the binding of the first three KDL molecules to MsbA. While we can't rule out that the resolved lipid bound states of MsbA represent binding of lipid to one or multiple site(s) on the transporter, the mole fraction plots are suggestive of binding to distinct sites, that is smooth inflections. In a previous study, (*Qiao et al., 2021*) we observed abnormal binding curves for some PIPs

**Table 6.** Thermodynamic signatures of KDL interacting with wild-type and mutant MsbA trapped with ADP and vanadate. Reported are the mean with standard deviation (n=3), and the subscript denotes the $n^{th}$ KDL binding event.

| | T (K) | $\Delta G_1$ (kJ/mol) | $\Delta H_1$ (kJ/mol) | $-T\Delta S_1$ (kJ/mol) | $\Delta Cp_1$ (kJ/mol/K) | $\Delta G_2$ (kJ/mol) | $\Delta H_2$ (kJ/mol) | $-T\Delta S_2$ (kJ/mol) | $\Delta Cp_2$ (kJ/mol/K) | $\Delta G_3$ (kJ/mol) | $\Delta H_3$ (kJ/mol) | $-T\Delta S_3$ (kJ/mol) | $\Delta Cp_3$ (kJ/mol/K) |
|---|---|---|---|---|---|---|---|---|---|---|---|---|---|
| WT | 293 | −35.3±0.1 | 21.9±1.5 | −57.2±1.3 | | −33.3±0.2 | 35.0±0.6 | −68.3±0.6 | | | | | |
| | 298 | −36.3±0.1 | 21.9±1.5 | −58.2±1.3 | | −34.4±0.2 | 35.0±0.6 | −69.4±0.6 | | | | | |
| | 303 | −37.3±0.1 | 21.9±1.5 | −59.2±1.3 | | −35.6±0.2 | 35.0±0.6 | −70.6±0.6 | | | | | |
| | 310 | −38.6±0.1 | 21.9±1.5 | −60.5±1.3 | | −37.2±0.2 | 35.0±0.6 | −72.2±0.6 | | | | | |
| R188A | 293 | −32.6±0.1 | −6.6±2.6 | −26.0±2.7 | 3.6±0.1 | −30.7±0.1 | −7.2±2.4 | −23.4±2.5 | 4.1±0.2 | −27.8±0.3 | −3.2±21.3 | −24.7±21.1 | 4.1±2.7 |
| | 298 | −33.2±0.2 | 11.3±2.6 | −44.5±2.7 | 3.5±0.1 | −31.2±0.1 | 13.4±3.3 | −44.7±3.4 | 4.2±0.3 | −28.5±0.4 | 17.6±12.4 | −46.1±12.3 | 3.6±4.2 |
| | 303 | −34.1±0.2 | 29.2±2.5 | −63.3±2.6 | 3.6±0.1 | −32.2±0.2 | 34.1±4.3 | −66.3±4.5 | 4.1±0.2 | −29.3±0.3 | 37.8±14.8 | −67.1±15.1 | 4.9±1.4 |
| | 310 | −35.8±0.2 | 54.3±2.6 | −90.2±2.8 | 3.6±0.1 | −34.0±0.3 | 63.0±5.8 | −97.0±6.1 | 4.1±0.2 | −31.3±0.5 | 67.9±26.1 | −99.2±26.6 | 4.3±2.2 |
| R238A | 293 | −32.1±0.2 | 126.8±5.8 | −158.9±5.9 | | −28.9±0.4 | 137.7±10.8 | −166.5±10.8 | | | | | |
| | 298 | −34.8±0.4 | 126.8±5.8 | −161.6±6.0 | | −31.7±0.4 | 137.7±10.8 | −169.4±11.0 | | | | | |
| | 303 | −37.6±0.4 | 126.8±5.8 | −164.3±6.1 | | −34.6±0.5 | 137.7±10.8 | −172.2±11.1 | | | | | |
| | 310 | −41.4±0.5 | 126.8±5.8 | −168.1±6.2 | | −38.5±0.7 | 137.7±10.8 | −176.2±11.4 | | | | | |
| K243A | 293 | −31.6±0.1 | 96.2±5.1 | −127.9±5.1 | | −29.1±0.2 | 111.7±6.9 | −140.8±6.6 | | | | | |
| | 298 | −33.8±0.1 | 96.2±5.1 | −130.0±5.1 | | −31.5±0.1 | 111.7±6.9 | −143.2±6.7 | | | | | |
| | 303 | −36.0±0.1 | 96.2±5.1 | −132.2±5.3 | | −33.9±0.1 | 111.7±6.9 | −145.6±6.9 | | | | | |
| | 310 | −39.1±0.2 | 96.2±5.1 | −135.3±5.4 | | −37.3±0.2 | 111.7±6.9 | −149±7.1 | | | | | |
| R188A R238A | 293 | −33.2±0.2 | −10.2±3.0 | −23.0±3.1 | 7.5±1.0 | −29.8±0.1 | −8.2±30.4 | −21.7±30.3 | 8.1±3.5 | | | | |
| | 298 | −33.9±0.2 | 27.2±2.5 | −61.0±2.6 | 7.5±0.3 | −30.5±0.5 | 31.9±13.7 | −62.4±14.2 | 8.6±4.6 | | | | |
| | 303 | −35.2±0.2 | 64.5±7.3 | −99.7±7.2 | 7.5±2.1 | −32.0±0.2 | 72.5±4.9 | −104.5±4.8 | 7.4±1.9 | | | | |
| | 310 | −38.1±0.4 | 116.9±16.4 | −155.0±16.8 | 7.5±1.3 | −35.0±0.2 | 127.7±25.9 | −162.7±25.9 | 7.9±3.0 | | | | |
| R188A K243A | 293 | −30.5±0.2 | 92.4±8.6 | −122.9±8.4 | −1.8±0.9 | −27.0±0.7 | 135.4±44.8 | −162.4±44.1 | −4.0±4.0 | | | | |
| | 298 | −32.3±0.1 | 82.1±4.4 | −114.4±4.3 | −0.1±0.9 | −29.5±0.1 | 114.7±25.2 | −144.2±25.2 | −3.4±4.5 | | | | |
| | 303 | −34.7±0.1 | 73.4±0.1 | −108.0±0.1 | −4.4±0.8 | −32.0±0.3 | 94.5±5.7 | −126.5±5.6 | −4.9±3.8 | | | | |
| | 310 | −36.7±0.1 | 55.6±5.7 | −92.3±5.7 | −2.6±0.8 | −34.5±0.1 | 64.4±22.9 | −98.9±22.9 | −4.3±3.9 | | | | |
| R238A K243A | 293 | −33.0±0.1 | 8.9±14.2 | −41.9±14.2 | 12.7±2.7 | −30.1±0.1 | 6.5±2.2 | −36.6±2.1 | 10.0±0.7 | | | | |
| | 298 | −34.2±0.2 | 71.8±7.2 | −106.0±7.4 | 13.2±2.1 | −31.0±0.1 | 56.1±2.1 | −87.1±2.2 | 10.5±0.9 | | | | |
| | 303 | −36.7±0.2 | 135.3±16.2 | −171.9±16.3 | 11.9±3.5 | −33.1±0.1 | 106.2±5.4 | −139.3±5.5 | 9.1±1.0 | | | | |
| | 310 | −41.6±0.8 | 222.4±35.8 | −264.0±36.5 | 12.4±2.9 | −37.0±0.3 | 174.4±10.5 | −211.4±10.8 | 9.7±0.7 | | | | |

**Table 7.** Double mutant cycle analysis of the first KDL binding to wild-type and mutant MsbA trapped with ADP and vanadate.
Shown as described in *Table 3*.

| | | Temperature (K) | ΔΔG (kJ/mol) | ΔΔH (kJ/mol) | Δ(-TΔS) (kJ/mol) | ΔΔG$_{int}$ (kJ/mol) | ΔΔH$_{int}$ (kJ/mol) | Δ(-ΔTS)$_{int}$ (kJ/mol) |
|---|---|---|---|---|---|---|---|---|
| R188A | | 293 | 2.7±0.1 | −28.5±3.1 | 31.2±3.1 | | | |
| | | 298 | 3.1±0.2 | −10.6±2.9 | 13.7±3.1 | | | |
| | | 303 | 3.2±0.2 | 7.3±2.9 | −4.1±2.9 | | | |
| | | 310 | 2.8±0.2 | 32.4±2.9 | −29.7±3.1 | | | |
| R238A | | 293 | 3.2±0.2 | 104.9±6.0 | −101.7±6.0 | | | |
| | | 298 | 1.5±0.4 | 104.9±6.0 | −103.4±6.1 | | | |
| | | 303 | −0.3±0.4 | 104.9±6.0 | −105.1±6.2 | | | |
| | | 310 | −2.8±0.5 | 104.9±6.0 | −107.6±6.4 | | | |
| K243A | | 293 | 3.7±0.1 | 74.3±5.4 | −70.7±5.3 | | | |
| | | 298 | 2.5±0.1 | 74.3±5.4 | −71.8±5.3 | | | |
| | | 303 | 1.3±0.1 | 74.3±5.4 | −73.0±5.4 | | | |
| | | 310 | −0.5±0.2 | 74.3±5.4 | −74.8±5.5 | | | |
| R188A | R238A | 293 | 2.2±0.2 | −32.1±3.3 | 34.2±3.4 | 3.8±0.4 | 108.5±7.0 | −104.7±7.2 |
| | | 298 | 2.4±0.2 | 5.3±2.9 | −2.8±2.9 | 2.2±0.5 | 89.0±6.9 | −86.9±7.1 |
| | | 303 | 2.1±0.2 | 42.6±7.5 | −40.5±7.3 | 0.8±0.5 | 69.6±9.6 | −68.7±9.8 |
| | | 310 | 0.5±0.4 | 95.0±16.5 | −94.5±16.8 | −0.5±0.6 | 42.4±17.6 | −42.8±18.1 |
| R188A | K243A | 293 | 4.8±0.2 | 70.5±8.7 | −65.7±8.6 | 1.6±0.2 | −24.7±10.4 | 26.3±10.3 |
| | | 298 | 4.0±0.1 | 60.2±4.7 | −56.2±4.5 | 1.6±0.2 | 3.5±7.2 | −1.9±7.2 |
| | | 303 | 2.6±0.1 | 51.5±1.5 | −48.8±1.3 | 1.9±0.2 | 30.1±5.8 | −28.2±5.9 |
| | | 310 | 1.9±0.1 | 33.7±5.9 | −31.8±5.9 | 0.4±0.4 | 73.0±8.1 | −72.7±8.3 |
| R238A | K243A | 293 | 2.3±0.1 | −13.0±14.3 | 15.3±14.2 | 4.6±0.2 | 192.2±16.2 | −187.7±16.2 |
| | | 298 | 2.1±0.2 | 49.9±7.3 | −47.8±7.6 | 1.9±0.4 | 129.3±10.5 | −127.4±10.9 |
| | | 303 | 0.6±0.2 | 113.4±16.3 | −112.7±16.4 | 0.4±0.5 | 65.8±17.9 | −65.4±18.2 |
| | | 310 | −3.0±0.9 | 200.5±35.8 | −203.5±36.5 | −0.4±1.0 | −21.3±36.6 | 21.1±37.5 |

binding to Kir3.2 that we rationalized by the presence of high-affinity binding and low-affinity binding sites. A revised equilibrium binding model including the two-site model dramatically improved the fits, leading to dissection of at least two lipid binding sites. Further studies are warranted to better understand the binding sites of KDL to MsbA in different conformations.

Results of this study begin to draw a connection between LPS binding at the interior and exterior sites of MsbA. It is presently thought that flipping of LOS occurs at interior MsbA site, and the exterior LOS binding site enables feedforward activation, wherein binding of LOS and precursors thereof stimulates ATPase activity (*Mi et al., 2017*; *Ho et al., 2018*; *Lyu et al., 2022*; *Gorzelak et al., 2021*; *Ward et al., 2007*; *Zou and McHaourab, 2009*; *Dong et al., 2005*). It is also thought that binding of LOS and ATP promotes dimerization of the NBDs. Here, we find mutations at either the interior or exterior sites have a direct impact of KDL binding to MsbA, which under these conditions is presumably adopting an open, IF conformation. Of the mutant proteins, MsbA containing single mutations (MsbA$^{R188A,K299A}$) at both LOS binding sites resulted in the greatest change in ΔG. This result implies that these sites are allosterically coupled and further investigation is warranted to better understand how the exterior LOS binding sites influence MsbA dynamics.

**Table 8.** Double mutant cycle analysis of the second KDL binding to wild-type and mutant MsbA trapped with ADP and vanadate.

Shown as described in *Table 3*.

| | | Temperature (K) | ΔΔG (kJ/mol) | ΔΔH (kJ/mol) | Δ(-TΔS) (kJ/mol) | ΔΔG$_{int}$ (kJ/mol) | ΔΔH$_{int}$ (kJ/mol) | Δ(-ΔTS)$_{int}$ (kJ/mol) |
|---|---|---|---|---|---|---|---|---|
| | | 293 | 2.7±0.2 | −42.2±2.4 | 44.9±2.6 | | | |
| | | 298 | 3.2±0.2 | −21.6±3.3 | 24.8±3.4 | | | |
| | | 303 | 3.5±0.4 | −0.9±4.4 | 4.3±4.5 | | | |
| R188A | | 310 | 3.2±0.4 | 28.0±5.9 | −24.8±6.1 | | | |
| | | 293 | 4.4±0.5 | 102.7±10.8 | −98.2±10.8 | | | |
| | | 298 | 2.7±0.5 | 102.7±10.8 | −100.0±11.0 | | | |
| | | 303 | 1.0±0.5 | 102.7±10.8 | −101.6±11.1 | | | |
| R238A | | 310 | −1.3±0.7 | 102.7±10.8 | −104.0±11.4 | | | |
| | | 293 | 4.2±0.4 | 76.7±6.9 | −72.5±6.6 | | | |
| | | 298 | 2.9±0.2 | 76.7±6.9 | −73.8±6.7 | | | |
| | | 303 | 1.7±0.2 | 76.7±6.9 | −75.0±6.9 | | | |
| K243A | | 310 | −0.1±0.4 | 76.7±6.9 | −76.8±7.1 | | | |
| | | 293 | 3.5±0.2 | −43.2±30.4 | 46.7±30.3 | 3.6±0.4 | 103.7±32.3 | −100.0±32.2 |
| | | 298 | 3.9±0.5 | −3.1±13.7 | 7.0±14.2 | 2.0±0.6 | 84.2±17.8 | −82.3±18.2 |
| | | 303 | 3.6±0.4 | 37.5±4.9 | −33.9±4.8 | 0.9±0.6 | 64.3±12.6 | −63.3±13.0 |
| R188A | R238A | 310 | 2.2±0.2 | 92.7±25.8 | −90.5±25.8 | −0.3±0.9 | 38.0±28.7 | −38.3±28.9 |
| | | 293 | 6.3±0.7 | 100.4±44.8 | −94.1±44.1 | 0.6±0.7 | −65.9±45.4 | 66.5±44.7 |
| | | 298 | 4.9±0.2 | 79.7±25.2 | −74.8±25.2 | 1.2±0.2 | −24.6±26.3 | 25.8±26.3 |
| | | 303 | 3.6±0.4 | 59.5±5.8 | −55.9±5.6 | 1.5±0.4 | 16.3±9.9 | −14.7±9.9 |
| R188A | K243A | 310 | 2.7±0.2 | 29.4±22.9 | −26.7±22.9 | 0.4±0.4 | 75.3±24.6 | −74.9±24.7 |
| | | 293 | 3.2±0.2 | −28.5±2.2 | 31.7±2.2 | 5.4±0.5 | 207.9±13.0 | −202.4±12.9 |
| | | 298 | 3.4±0.2 | 21.1±2.2 | −17.7±2.3 | 2.2±0.4 | 158.3±13.0 | −156.1±13.1 |
| | | 303 | 2.5±0.2 | 71.2±5.4 | −68.7±5.5 | 0.2±0.5 | 108.2±13.8 | −107.9±14.2 |
| R238A | K243A | 310 | 0.2±0.4 | 139.4±10.5 | −139.2±10.8 | −1.6±0.9 | 40.0±16.5 | −41.6±17.3 |

A defining feature of this work is the use of mutant cycles to not only characterize specific membrane protein-lipid interactions but define the coupling energies of specific residue-lipid interactions in terms of enthalpic and entropic contributions. Traditionally, mutant cycles have been used to understand pairwise interactions of residues, such as in protein-protein complexes, in terms of coupling free energy. Here, we extend mutant cycles to understand how pairs of residues contribute to specific MsbA-KDL interactions. Double mutants targeting the interior site reveal a positive coupling energy of nearly 2 kJ/mol for R78 and K299. These stabilize the MsbA-KDL complex largely through nearly 17 kJ/mol of favorable coupling entropy, which outweighs a negative coupling enthalpy. This phenomenon extends to nearly all mutant cycles investigated in this work, even when the transporter is trapped with vanadate. The largest coupling energy is observed from the triple mutant cycle of R188A, R238A, and R243A, which again stabilization of the complex was achieved via favorable coupling entropy. While we focused on results at 298 K, the coupling energetics among these three residues show 3.4±0.5 kJ/mol. Taken together, mutant cycle analysis reveals that entropy drives high-affinity KDL binding to MsbA and solvent reorganization contributes to KDL binding (*Figure 6*). There are many factors that contribute to the change in entropy of the system, beyond solvation entropy, and deciphering the entropic contributions of the various components warrants additional studies.

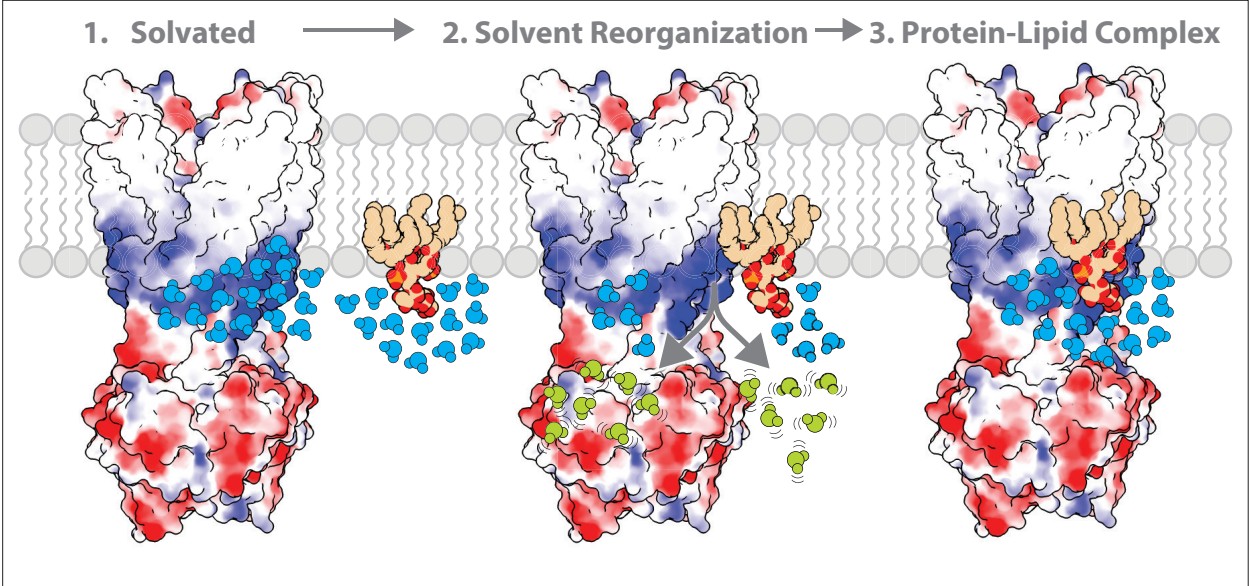

**Figure 6.** The role of solvent in contributing to the molecular recognition of membrane protein-lipid complexes. The lipid headgroup and binding pocket (basic patch illustrated in blue) on the membrane protein are solvated. The ordered solvent (shown in light blue) is then displaced upon lipid binding the membrane protein leading to solvent reorganization. The displacement of ordered solvent (show in light green) contributes to favorable entropy. This process enables the formation of a high affinity, stable membrane protein-lipid complex.

While the use of mutant cycles was prominent a few decades ago, native mass spectrometry opens new opportunities to revisit the classical approach, diving deeper into the energetics of non-covalent interactions, such as dissecting energetics in terms of enthalpic and entropic contributions. Native MS coupled with a variable-temperature nanoelectrospray ionization (nESI) apparatus (*McCabe et al., 2021*; *Cong et al., 2016*) has been used to ascertain equilibrium binding constants and thermodynamic properties of protein-protein and protein-ligand interactions. The results obtained align closely with those obtained through other biophysical techniques, such as isothermal titration calorimetry (ITC) and surface plasmon resonance (SPR) (*Cong et al., 2017*; *Daneshfar et al., 2004*; *Daneshfar et al., 2004*; *Cong et al., 2016*). The approach has also uncovered that specific protein-lipid interactions can allosterically modulate other interactions with protein, lipid, and drug molecules (*Marcoux et al., 2013*; *Yen et al., 2018*; *Cong et al., 2017*; *Patrick et al., 2018*; *Bolla et al., 2018*; *Gault et al., 2016*). More recently, native MS has proved useful in dissecting the thermodynamics of individual nucleotide binding events to GroEL, a 801 kDa tetradecameric chaperonin (*Walker et al., 2023*). In contrast to traditional approaches, such as ITC and SPR, native MS can resolve and dissect individual binding events enabling the measurement of binding thermodynamics, which is of paramount importance to understanding the molecular driving forces of non-covalent interactions.

In summary, we demonstrate the utility of native MS to determine the thermodynamic origins of specific KDL-MsbA interactions. Combined with the classical mutant cycle approach, (*Carter et al., 1984*) the thermodynamic contribution of specific interactions with lipids is illuminated. More specifically, MsbA binding KDL is solely driven by entropy, which overcomes an enthalpic penalty. A similar thermodynamic strategy was also observed for Kir3.2-PIP interactions, where entropy plays a central role in the wild-type channel recognizing PIP (*Qiao et al., 2021*). It is tempting to speculate that favorable entropy is a common theme enabling membrane proteins to specifically engage carbohydrate-containing lipids. We envision thermodynamics and mutant cycles will be invaluable in not only better understanding high-affinity lipid binding sites but also in the development of inhibitors, such as those that may target specific protein-lipid binding site(s). In closing, these studies provide deeper insight into the thermodynamic strategies membrane proteins exploit to achieve high-affinity lipid binding site(s).

# Materials and methods

## Key resources table

| Reagent type (species) or resource | Designation | Source or reference | Identifiers | Additional information |
|---|---|---|---|---|
| Strain, strain background (*Escherichia coli*) | BL21-AI | Invitrogen | C607003 | Chemically Competent *E. coli* |
| Strain, strain background (*Escherichia coli*) | 5-alpha | NEB | C2987H | Chemically Competent *E. coli* |
| Recombinant DNA reagent | pCDF-His_TEV_MsbA (plasmid) | DOI: 10.1038 /s41467-022-34905-2 | | MsbA expression construct |
| Sequence-based reagent | MsbA_R78A_F | This Paper | PCR primers | gcgGGTATCACCAGCTATGTC |
| Sequence-based reagent | MsbA_R78A_R | This Paper | PCR primers | CAAAATCATCAGCCCGATC |
| Sequence-based reagent | MsbA_R188A_F | DOI: 10.1038 /s41467-022-34905-2 | PCR primers | gcgTTTCGCAACATCAGTAAAAAC |
| Sequence-based reagent | MsbA_R188A_R | DOI: 10.1038 /s41467-022-34905-2 | PCR primers | CTTCGATACTACGCGAATC |
| Sequence-based reagent | MsbA_R238A_F | DOI: 10.1038 /s41467-022-34905-2 | PCR primers | gcgCTTCAGGGGATGAAAATG |
| Sequence-based reagent | MsbA_R238A_R | DOI: 10.1038 /s41467-022-34905-2 | PCR primers | CATTCGGTTGCTGACTTTATC |
| Sequence-based reagent | MsbA_K243A_F | DOI: 10.1038 /s41467-022-34905-2 | PCR primers | gcgATGGTTTCAGCCTCTTCC |
| Sequence-based reagent | MsbA_K243A_R | DOI: 10.1038 /s41467-022-34905-2 | PCR primers | CATCCCCTGAAGACGCAT |
| Sequence-based reagent | MsbA_K299A_F | This Paper | PCR primers | gcgTCGCTGACTAACGTTAACGC |
| Sequence-based reagent | MsbA_K299A_R | This Paper | PCR primers | CAGCGGACGCATCAGTGC |
| Commercial assay or kit | DC Protein Assay | Bio-Rad | 5000112 | |
| Chemical compound, drug | Kdo2-Lipid A (KLA) | Avanti | 699500 | |

## MsbA expression constructs

MsbA and mutants were essentially expressed and purified as previously described (*Lyu et al., 2022*). In detail, the MsbA gene (from *Escherichia coli* genomic DNA) was amplified by polymerase chain reaction (PCR) using Q5 High-Fidelity DNA Polymerase (New England Biolabs, NEB) and subcloned into a modified pCDF-1b plasmid (Novagen) resulting in expression of MsbA with an N-terminal TEV protease cleavable His6 fusion protein. Primers for generating mutations for MsbA were designed using the online tool NEBaseChanger (NEB) and carried out using the KLD enzyme mix (NEB) as described by the manufacturer. All plasmids were transformed into *E. coli* 5-alpha (NEB) competent cells for plasmids propagation before confirmed by DNA sequencing.

## Protein expression and purification

MsbA expression plasmids were transformed into *E. coli* (DE3) BL21-AI competent cells (Invitrogen). A single colony was picked and used to inoculate 50 mL LB media to be grown overnight at 37 °C with shaking. The overnight culture was used to inoculate to terrific broth (TB) media and incubated at 37 °C until the $OD_{600nm} \approx 0.6$–1.0. After which, the cultures were induced with final concentration of 0.5 mM IPTG (isopropyl β-D-1-thiogalactopryanoside) and 0.2% (w/v) arabinose. After overnight expression at 25 °C, the cultures were harvested at 4000 x g for 10 minutes and the resulting pellet was resuspended in lysis buffer (20 mM Tris, 300 mM NaCl and pH at 7.4 at room temperature). The resuspended cells were centrifuged, and the pellet was then resuspended in lysis buffer. Cells were lysed by four passages through a Microfluidics M-110P microfluidizer operating at 25,000 psi with reaction chamber emersed in an ice bath. The lysate was clarified by centrifugation at 20,000 x g for 25 min and the supernatant was centrifuged at 100,000 x g for 2 hr to pellet membranes. Resuspension buffer (20 mM Tris, 150 mM NaCl, 20% (v/v) glycerol, pH 7.4) was used to homogenize the resulting pellet and 1% (m/v) DDM was added for protein extraction overnight at 4 °C. The

extraction was centrifuged at 20,000 x $g$ for 25 min and the resulting supernatant was supplemented with 10 mM imidazole and filtered with a 0.45 µm syringe filter prior to purification by immobilized metal affinity chromatography. The extraction containing solubilized MsbA was loaded onto a column packed with 2.5 mL Ni-NTA resin pre-equilibrated in NHA-DDM buffer (20 mM Tris, 150 mM NaCl, 10 mM imidazole, 10% (v/v) glycerol, pH 7.4 and supplemented with 2 x the critical micelle concentration (CMC) of DDM). After the loading, the column was washed with 5 column volumes (CV) of NHA-DDM buffer, 10 CV of NHA-DDM buffer supplemented with additional 2% (w/v) nonyl-ß-glucoside (NG), and 5 CV of NHA-DDM buffer. The immobilized protein was eluted with the addition of 2 CV of NHB-DDM buffer (20 mM Tris, 150 mM NaCl, 250 mM imidazole, 10% (v/v) glycerol, 2 x CMC of DDM, pH 7.4). The eluted MsbA was pooled and desalted using HiPrep 26/10 desalting column (GE Healthcare) pre-equilibrated in desalting buffer (NHA-DDM with imidazole omitted). TEV protease (expressed and purified in-house) was added to the desalted MsbA sample and incubated overnight at room temperature. The sample was passed over a pre-equilibrated Ni-NTA column and the flow-through containing the cleaved MsbA protein was collected. The pooled protein was concentrated using a centrifugal concentrator (Millipore, 100 kDa) prior to injection onto a Superdex 200 Increase 10/300 GL (GE Healthcare) column equilibrated with 20 mM Tris, 150 mM NaCl, 10% (v/v) glycerol and 2 x CMC $C_{10}E_5$. Peak fractions containing dimeric MsbA were pooled, flash frozen in liquid nitrogen, and stored at –80 °C prior to use.

## Preparation of MsbA for native MS studies

MsbA samples were incubated with 20 µM copper (II) acetate, to saturate the N-terminal metal binding site, (*Lyu et al., 2022*) prior to buffer exchange using a centrifugal buffere exchange device (Bio-Spin, Bio-Rad) into 200 mM ammonium acetate supplemented with 2 x CMC of $C_{10}E_5$. To prepare vanadate-trapped MsbA, ATP and $MgCl_2$ were added to MsbA at a final concentration of 10 mM. After incubation at room temperature for 10 min, a freshly boiled vanadate solution (pH 10) was added to reach final concentration of 1 mM followed by incubation at 37 °C for an additional 10 min. The sample was then buffer exchanged as described above.

## Native Mass Spectrometry

Samples were loaded into gold-coated glass capillaries made in-house (*Laganowsky et al., 2013*) and introduced into at a Thermo Fisher Scientific Exactive Plus Orbitrap with Extended Mass Range (EMR) using native electrospray ionization source modified with a variable temperature apparatus (*McCabe et al., 2021*). For native mass analysis, the instrument was tuned as follow: source DC offset of 10 V, injection flatapole DC to 8.0 V, inter flatapole lens to 4, bent flatapole DC to 3, transfer multipole DC to 3 and C trap entrance lens to 0, trapping gas pressure to 6.0 with the in-source CID to 65.0 eV and CE to 100, spray voltage to 1.70 kV, capillary temperature to 200 °C, maximum inject time to 200ms. Mass spectra were acquired with a setting of 17,500 resolution, microscans set to 1 and averaging set to 100.

## Determination MsbA-lipid equilibrium binding constants

KDL (Avanti) stock solution was prepared by dissolving lipid powder in water. The concentration of MsbA and KDL were determined by a DC protein assay (BioRad) and phosphorus assay, respectively (*Chen et al., 1956*; *Fiske and Subbarow, 1925*). MsbA was incubated with varying concentrations of KDL before loading into a glass emitter and mounted on a variable-temperature electrospray ionization (vT-ESI) source (*McCabe et al., 2021*). Samples were incubated in the source for two minutes at the desired temperature before data acquisition. All titration data were collected in triplicate ($n=3$), with a 20 min interval between each measurement. Reported are the mean and standard deviation. At a given temperature, the mass spectra were deconvoluted using Unidec (*Marty et al., 2015*) and the peak intensities for apo and KDL-bound species were determined and converted to mole fraction. The sequential ligand binding model was applied to determine the mole fraction of each species in measurement:

$$PL_{n-1} + L \overset{K_A}{\Leftrightarrow} PL_n$$

where:

$$K_{An} = \frac{[PL_n]}{[PL_{n-1}][L]}$$

To calculate the mole fraction of a particular species (**Cong et al., 2016**):

$$F_{PLn} = \frac{[L]_{free}^n \prod_{j=1}^n K_{Aj}}{1 + \sum_{i=1}^n [L]_{free}^i \prod_{j=1}^n K_{Aj}}$$

For each titrant in the titration, the free concentration of lipid was computed as follows:

$$[L]_{free} = [L]_{total} - [P]_{total} \sum_{i=0}^n i F_{PLi}$$

The sequential ligand binding model was globally fit to the mole fraction data by minimization of pseudo-$\chi^2$ function:

$$\chi^2 = \sum_{j=1}^m \sum_{k=1}^d \left(F_{i,j,exp} - F_{i,j,calc}\right)^2$$

where $n$ is the number of bound ligands and $d$ is the number of the experimental mole fraction data points.

Van't Hoff analysis (**van't Hoff, 1884**) was applied to determine the Gibbs free energy change (ΔG), enthalpy change (ΔH) and entropy change (ΔS) based on the equation:

$$\ln K_A = -\frac{\Delta H}{R} \cdot \frac{1}{T} + \frac{\Delta S}{R}$$

For non-linear trends, the non-linear form of the Van't Hoff equation was applied to determine the thermodynamic parameters (**Liu and Sturtevant, 1996**):

$$\ln K_A = \frac{\Delta H_{T_0} - T_0 \Delta C_p}{R}\left(\frac{1}{T_0} - \frac{1}{T}\right) + \frac{\Delta C_p}{R} \ln\left(\frac{T}{T_0}\right) + \ln K_0$$

where $K_A$ is the equilibrium association constant, $K_0$ is the equilibrium association constant at the reference temperature ($T_0$), $\Delta H_{T_0}$ is the standard enthalpy at $T_0$, $\Delta C_p$ is the change in heat capacity at constant pressure, and R is the universal gas constant.

## Mutant cycle analysis

If the two mutated residues are interacting, then the coupling free energy (ΔΔG_int) will not be 0 and the value may be positive or negative depending upon whether the interactions between mutated residues enhance or weaken the functional property measured. (**Wells, 1990**) ΔΔG_int can be computed given the change in Gibbs free energy for the wild-type protein (P), two single mutants (PX and PY), and double mutant (PXY) as follows:

$$\Delta\Delta G_{int} = \Delta\Delta G_{PX \to P} + \Delta\Delta G_{PY \to P} - \Delta\Delta G_{PXY \to P}$$

where $\Delta\Delta G_{PX \to P} = \Delta G_{PX} - \Delta G_P$, $\Delta\Delta G_{PY \to P} = \Delta G_{PY} - \Delta G_P$ and $\Delta\Delta G_{PXY \to P} = \Delta G_{PXY} - \Delta G_P$. Analogously, the contributions from coupling enthalpy (ΔΔH_int) and coupling entropy (Δ(-TΔS_int)) can be computed as follows:

$$\Delta\Delta H_{int} = \Delta\Delta H_{PX \to P} + \Delta\Delta H_{PY \to P} - \Delta\Delta H_{PXY \to P}$$

$$\Delta\left(-T\Delta S_{int}\right) = \Delta\left(-T\Delta S_{PX \to P}\right) + \Delta\left(-T\Delta S_{PY \to P}\right) - \Delta\left(-T\Delta S_{PXY \to P}\right)$$

where T is temperature in K. As an example, $\Delta\Delta H_{PX \to P} = \Delta H_{PX} - \Delta H_P$ and $\Delta\left(-T\Delta G_{PX \to P}\right) = T\Delta S_P - T\Delta S_{PX}$.

## Acknowledgements

This work was supported by National Institutes of Health (NIH) under grant numbers (DP2GM123486, R01GM121751, R01GM139876, R01GM138863 and RM1GM145416 to AL; P41GM128577 to DR; and R35GM128624 to MM).

## Additional information

### Funding

| Funder | Grant reference number | Author |
| --- | --- | --- |
| National Institute of General Medical Sciences | R01GM139876 | Arthur Laganowsky |
| National Institute of General Medical Sciences | R01GM138863 | David H Russell |
| National Institute of General Medical Sciences | RM1GM149374 | David H Russell |
| National Institute of General Medical Sciences | R35GM128624 | Michael T Marty |
| National Institute of General Medical Sciences | RM1GM145416 | Arthur Laganowsky |
| National Institutes of Health | DP2GM123486 | Arthur Laganowsky |
| National Institutes of Health | R01GM121751 | Arthur Laganowsky |
| National Institutes of Health | P41GM128577 | Arthur Laganowsky David H Russell |
| National Institutes of Health | R35GM128624 | Michael T Marty |

The funders had no role in study design, data collection and interpretation, or the decision to submit the work for publication.

### Author contributions

Jixing Lyu, Conceptualization, Data curation, Formal analysis, Investigation, Writing – original draft; Tianqi Zhang, Formal analysis, Investigation, Writing – review and editing; Michael T Marty, David Clemmer, Formal analysis, Writing – review and editing; David H Russell, Formal analysis, Funding acquisition, Writing – review and editing; Arthur Laganowsky, Conceptualization, Software, Formal analysis, Supervision, Funding acquisition, Writing – original draft, Project administration

### Author ORCIDs

Michael T Marty ⓘ http://orcid.org/0000-0001-8115-1772
Arthur Laganowsky ⓘ https://orcid.org/0000-0001-5012-5547

Reviewer #1 (Public Review): https://doi.org/10.7554/eLife.91094.3.sa1
Reviewer #3 (Public Review): https://doi.org/10.7554/eLife.91094.3.sa2
Author Response https://doi.org/10.7554/eLife.91094.3.sa3

## Additional files

### Supplementary files
• MDAR checklist

### Data availability
Source data has been deposited at Zenodo (https://zenodo.org/records/10403301).

The following dataset was generated:

| Author(s) | Year | Dataset title | Dataset URL | Database and Identifier |
|---|---|---|---|---|
| Lyu J, Zhang T, Marty MT, Clemmer D, Russell DH, Laganowsky A | 2023 | Double and triple thermodynamic mutant cycles reveal the basis for specific MsbA-lipid interactions | https://doi.org/ 10.5281/zenodo. 10403301 | Zenodo, 10.5281/ zenodo.10403301 |

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
