## [Editor Report · eLife assessment]

This is an **important** biophysical study combining native mass spectrometry with mutant cycles to estimate the thermodynamic components of lipid A binding to the ABC transporter MsbA. **Solid** evidence supports the binding energies for lipid-protein interactions to MsbA using this approach, which could be later applied to other membrane proteins in general.

---

## [Referee Report · Reviewer #1 (Public Review)]

Summary:

The preprint by Laganowsky and co-workers describes the use of mutant cycles to dissect the thermodynamic profile of specific lipid recognition by the ABC transporter MsbA. The authors use native mass spectrometry with a variable temperature source to monitor lipid binding to the native protein dimer solubilized in detergent. Analysis of the peak intensities (that is, relative abundance) of 1-3 bound lipids as a function of solution temperature and lipid concentration yields temperature-dependent Kds. The authors use these to then generate van't Hoff plots, from which they calculate the enthalpy and entropy contributions to binding of one, two, and in some cases, three lipids to MsbA. The authors have previously demonstrated that MS can indeed extract thermodynamic contributions to lipid binding. The authors then employ mutant cycles, in which basic residues involved in headgroup binding are mutated to alanine. By comparing the thermodynamic signatures of single and double (and in one instance triple) mutants, they aim to identify cooperativity between the different positions. They furthermore use inward and outward locking conditions which should control access to the different binding sites determined previously. The main conclusion is that lipid binding to MsbA is driven mainly by energetically favorable entropy increase upon binding, which stems from the release of ordered water molecules that normally coordinate the basic residues, which helps to overcome the enthalpic barrier of lipid binding. The authors also report an increase in lipid binding at higher temperatures which they attribute to a non-uniform heat capacity of the protein. Although they find that most residue pairs display some degree of cooperativity, particularly between the inner and outer lipid binding sites, they do not provide a structural interpretation of these results.

Strengths:

The use of double mutant cycles and mass spectrometry to dissect lipid binding is novel and interesting. For example, the observation that mutating a basic residue in the inner and one in the outer binding site abolishes lipid binding to a greater extent than the individual mutations is highly informative even without having to break it down into thermodynamic terms. The method and data reported here opens new avenues for the structure/activity relationship of MsbA. The "mutant cycle" approach is in principle widely applicable to other membrane proteins with complex lipid interactions.

Weaknesses:

The use of double mutant cycles to dissect binding energies is well-established, and has, as the authors point out, been employed in combination with mass spectrometry to study protein-protein interactions. Its application to extract thermodynamic parameters is robust in cases where a single binding event is monitored, e.g. the formation of a complex with well-defined stoichiometry, where dissociation constants can be determined with high confidence. It is, however, complicated significantly by the fact that for MsbA-lipid interactions, we are not looking at a single binding event, but a stochastic distribution of lipids across different sites. Even if the protein is locked in a specific conformation, the observation of a single lipid adduct does not guarantee that the one lipid is always bound to a specific site. The authors discuss this issue in the manuscript. As they point out, one can assume that the most high-affinity sites will be populated first. Hence, the Kd values determined by MS likely describe (mostly) lipid binding to these sites, although this does not seem to hold universally true, as seen for example for the two (in principle equivalent) binding sites in the vanadate-locked protein. In addition, mutation of a binding site (which the authors show reduces lipid binding) may instead allow the lipid to bind to a lower-affinity site elsewhere. In summary, the Kds are an approximation.

(Minor comment: The protein concentrations used for MS titration experiments should be stated in the methods.)

The authors conclude that solvation entropy is a major factor driving lipid binding (Figure 6). If the increase in entropy upon lipid binding comes from the release of ordered water molecules around the basic residues, we should see a smaller increase in entropy for proteins where several basic residues have been changed to alanine, which is not the case. The authors explain this by stating that other entropic factors likely are at play. Judging from their data, that is certainly correct, but why then focus on solvation entropy in the discussion if its contribution to the total entropy change cannot be determined?

---

## [Referee Report · Reviewer #3 (Public Review)]

Summary:

In this paper presented by Liu et al, native MS on the lipid A transporter MsbA was used to obtain thermodynamic insight into protein-lipid interactions. By performing the analyses at different lipid A concentrations and temperatures, dissociation constants for 2-3 lipid A binding sites were determined, as well as enthalpies were calculated using non-linear van't Hoff fitting.

Strengths:

This is an extensive high quality native MS dataset that provides unique opportunities to gain insights into the thermodynamic parameters underlying lipid A binding. In addition, it provides coupling energies between mutations introduced into MsbA, that are implicated in lipid A binding.

Weaknesses:

It remains elusive, which KD values belong to which of the possible lipid A binding sites.

Appraisal:

The authors convincingly addressed the concerns raised by the reviewers.

---

## [Author Response]

The following is the authors’ response to the original reviews.

We thank the reviewers for their careful, critical, and insightful evaluation of our manuscript.

**Public Reviews:**

**Reviewer #1 (Public Review):**
Summary:The preprint by Laganowsky and co-workers describes the use of mutant cycles to dissect the thermodynamic profile of specific lipid recognition by the ABC transporter MsbA. The authors use native mass spectrometry with a variable temperature source to monitor lipid binding to the native protein dimer solubilized in detergent. Analysis of the peak intensities (that is, relative abundance) of 1-3 bound lipids as a function of solution temperature and lipid concentration yields temperature-dependent Kds. The authors use these to then generate van't Hoff plots, from which they calculate the enthalpy and entropy contributions to binding of one, two, and in some cases, three lipids to MsbA.The authors then employ mutant cycles, in which basic residues involved in headgroup binding are mutated to alanine. By comparing the thermodynamic signatures of single and double (and in one instance triple) mutants, they aim to identify cooperativity between the different positions. They furthermore use inward and outward locking conditions which should control access to the different binding sites determined previously.The main conclusion is that lipid binding to MsbA is driven mainly by energetically favorable entropy increase upon binding, which stems from the release of ordered water molecules that normally coordinate the basic residues, which helps to overcome the enthalpic barrier of lipid binding. The authors also report an increase in lipid binding at higher temperatures which they attribute to a non-uniform heat capacity of the protein. Although they find that most residue pairs display some degree of cooperativity, particularly between the inner and outer lipid binding sites, they do not provide a structural interpretation of these results.Strengths:The use of double mutant cycles and mass spectrometry to dissect lipid binding is novel and interesting. For example, the observation that mutating a basic residue in the inner and one in the outer binding site abolishes lipid binding to a greater extent than the individual mutations is highly informative even without having to break it down into thermodynamic terms (see "weaknesses" section). In this sense, the method and data reported here opens new avenues for the structure/activity relationship of MsbA. The "mutant cycle" approach is in principle widely applicable to other membrane proteins with complex lipid interactions.Weaknesses:The use of double mutant cycles to dissect binding energies is well-established, and has, as the authors point out, been employed in combination with mass spectrometry to study protein-protein interactions. Its application to extract thermodynamic parameters is robust in cases where a single binding event is monitored, e.g. the formation of a complex with well-defined stoichiometry, where dissociation constants can be determined with high confidence. It is, however, complicated significantly by the fact that for MsbA-lipid interactions, we are not looking at a single binding event, but a stochastic distribution of lipids across different sites. Even if the protein is locked in a specific conformation, the observation of a single lipid adduct does not guarantee that the one lipid is always bound to a specific site. In some of the complexes detected by MS, the lipid is likely bound somewhere else. Lipid binding Kds from mass spectrometry, although helpful in some instances as a proxy for global binding affinities, should therefore be taken with a grain of salt.

We agree with the reviewer in that while we will measure binding of lipid (mass shift) we do not know the binding location(s). Given this issue, we have added to the discussion section on this important point and elaborate more broadly on this problem in the context of membrane protein-lipid interactions. Tackling this issue represents a frontier challenge for the field.

The authors analyze the difference in binding upon mutating binding sites (ddG etc). Here, another complicating factor comes into play, the fact that mutation of a binding site (which the authors show reduces lipid binding) may instead allow the lipid to bind to a lower-affinity site elsewhere. Unfortunately, the authors do not specify the protein concentration, but assuming it is in the single-digit micromolar range, as common for native MS experiments, lipid and protein concentrations are almost equal for most of the data points, resulting in competition between binding sites for free lipids. As a rule of thumb, for Kd measurements, the concentration of the constant component, the protein, should be far below the Kd, to avoid working in the "titration" regime rather than the "binding" regime (see Jarmoskaite et al, eLife 2020). I cannot determine whether this is the case here. The way I understand the double mutant cycle approach, reliable Kd measurements are required to accurately determine dH and TdS, so I would encourage the authors to confirm their Kd values using complementary methods before in-depth interpretations of the thermodynamic components.

The reviewer references an article in eLife by Jarmoskaite and co-workers describing “titration” vs “binding” regimes. Below we paste a snippet from this article:

Equation 4a is an expression for the fraction of protein bound to ligand, which universally holds, i.e., if we know the concentration of molecules at equilibrium (including those unbound or free) then one can obtain the special ratio or equilibrium constant at a given temperature. Jarmoskaite et al. note that in practice (using traditional biophysical approaches) one cannot readily distinguish protein that is free or bound to ligand (see highlighted part above). While this assumption is basis of their eLife assessment, it does NOT apply to native mass spectrometry data. It is important to realize that the mole fraction (or concentration) of apo and each lipid bound states, i.e., [P], [PL], [PL2], …, [PLn+1], can readily be obtained directly from the deconvoluted mass spectrum. This is unlike other biophysical methods that are ensemble measurements, which measures the amount of heat or fraction of total ligand bound to protein. Since we can discern each lipid bound state, including the free protein and free ligand concentrations, the equilibrium binding constants can be directly calculated, and the protein and ligand concentration becomes irrelevant. In principle, equilibrium constants for protein-lipid interactions can be calculated from one mass spectrum. To increase transparency, we have updated the results section to highlight the important difference of the native MS approach compared to less robust traditional approaches that are riddled with underlying issues/assumptions.

We appreciated the reviewer’s suggestion of using complementary methods to confirm Kd values. In our previous report [1], we determined binding thermodynamics for soluble protein-ligand interactions using native MS, surface plasmon resonance (SPR), and isothermal calorimetry (ITC) and found the techniques yield similar binding constants and thermodynamic parameters. The use of soluble proteins with defined ligand binding studies was rather straightforward to carry out a complementary study. We have also shown consistent findings for native MS and SPR of membrane protein interaction with a soluble, regulatory protein [2]. However, in the case of membrane proteins they can bind the first few lipids very specifically and, with the addition of more lipid, bind even more lipids that represent rather weak binding. Thus, traditional approaches would report on the ensemble of lipids bound to membranes and specific lipid binding sites (such as inner and outer LPS binding sites in MsbA) are saturable but also additional binding will be observed, i.e., doesn’t follow traditional soluble protein-ligand binding studies. In the past we have used a fluorescent-lipid competition binding assay [3] to corroborate native MS results for Kir3.2, which showed a direct correlation. The disadvantage of this complementary approach is using a non-natural, fluorescent-modified lipid. Unfortunately, there is no commercial source for a fluorophore modified KDL.

It is somewhat counterintuitive that for many double mutants, and the triple mutant, the entropic component becomes more favorable compared to the WT protein. If the increase in entropy upon lipid binding comes from the release of ordered water molecules around the basic residues (a reasonable assumption) why does this apply even more in proteins where several basic residues have been changed to alanine, which coordinate far fewer water molecules?

There are many factors that contribute to the change in entropy of the system, beyond solvation entropy, and deciphering the entropic contributions of the various components remains a challenging task. We have revised the manuscript to emphasize that solvation is one component of the entropic term and other components are likely at play.

The authors could devote more attention to the fact that they use detergent micelles as a vehicle for lipid binding studies. To a limited extent, detergents compete with lipids for binding, and are present in extreme excess over the lipid. The micelle likely changes its behavior in response to temperature changes. For example, the packing around the protein loosens up upon heating, which may increase the chance for lipids to bind. In this case, the increase in binding at higher temperatures may not be related to a change in heat capacity. This question could be addressed by MD simulations, if it's not already in the literature.

The detergent and its concentration are consistent for all the different MsbA proteins in this study. In fact, we observe linear van’t Hoff plots with positive and negative slopes as well as non-linear curves that are convex or concave. The MsbA protein (wt or mutant), trapped or not, all display unique temperature-dependent responses. The reviewers comment of increasing temperature to loosen packing of detergent to promote lipid binding is clearly NOT that simple. If detergent was significantly influencing lipid binding (as suggested by reviewer) then increasing its concentration should impact lipid binding. In a previous study, we found no difference in membrane protein-lipid thermodynamics even when the concentration of detergent was increased five-fold [1]. We repeated similar experiments for MsbA and find the increased detergent concentration does not impact the abundances of lipid bound states. The figure to the right shows MsbA in the presence of lipid in 2x CMC (panel a and b) and 10x CMC (panel c and d). As you will see, no appreciably difference in the lipid bound signal is observed.

**Author response image 2. sa3fig2:** 

We applaud the suggestion of MD simulation. However, it is far beyond the scope of this paper and its not clear what will really be learned.

**Reviewer #2 (Public Review):**
Summary:This is a solid study that dissects the thermodynamics of lipopolysaccharide (LPS) transporter MsbA and LPS. Native ESI-MS and the novel strategies developed by the authors were employed to quantify the affinities of LPS-MsbA interactions and its temperature dependence. Here, the equilibrium of lipid-protein interactions occurs in the micellar phase. The double-/triple-mutant cycle analysis and van't Hoff analysis allowed a full thermodynamic description of the lipid-protein interactions and the analysis of thermodynamic coupling between LPS binding sites. The most notable result would be that LPS-MsbA interaction is largely driven by entropy involving the negative heat capacity, a signature of the solvent reorganization effect (here authors attribute the solvent effect to "water" reorganization). The entropy driven lipid binding has been previously reported by the same authors for Kir1,2-PIP2 interactions.Strengths:1. This is overall a very thorough and rigorous study providing the detailed thermodynamic principles of LPS-MsbA interaction.1. The double and triple-mutant cycle approaches are newly applied to lipid-protein interactions, enabling detailed thermodynamics between LPS binding sites.1. The entropy-driven protein-lipid interaction is surprising. The binding seems to be mainly mediated by the electrostatic interaction between the positively charged residues on the protein and the negatively charged or polar headgroup of LPS, which could be thought of as "enthalpic" (making of a strong bond relative to that with solvent).Weaknesses:

1. This study is a good contribution to the field, but it was difficult to find novel biological insights or methodological novelty from this study.

1a. Thermodynamic analysis of lipid-protein interactions, an example of entropy-driven lipid-protein interactions, and the cooperativity between lipid binding sites have been reported by the author's group. Also, the cooperativity between binding sites in general have been reported from numerous studies of biomolecular interactions.

We appreciate the reviewer for highlighting our previous work. Of course, a single study does not establish a pattern, such as entropy-driven lipid-protein interactions.

While we agree with the reviewer that cooperativity in biomolecular interactions has been established for many soluble protein systems, by no means do we have a detailed understanding of membrane protein-lipid interactions. This work is an important contribution to expanding on classical work on soluble protein systems to more challenging membrane protein systems and their interactions with lipids.

1b. It is not clear how this study provides new insights into the understanding of LPS transport mechanisms. Probably, authors could strengthen the Discussion by providing biological insights-how the residue coupling.

The thermodynamics provides us with a deeper insight into the chemical principles that drive specific membrane protein-lipid interactions. We have revised the discussion to highlight the importance of thermodynamics and the implication of individual residues to KDL binding, and the inner and outer LPS binding sites appear to be coupled, something that is new.

1. One to three LPS molecules bind to MsbA, but it is unclear whether bound KDL occupies inner or outer cavities, or both and how a specific mutation affects the affinity of specific LPS (i.e., to inner or to outer cavities). Based on the known structures, the maximal number of LPS is three. It is possible that the inner and outer cavities have different LPS affinities. Also, there can be multiple one-LPS-bound states, two-LPS-bound states if LPS strictly binds to the binding sites indicated by the structures. This aspect is beyond the scope of this study and difficult to address, but without this information, it seems hard to tell what is going on in the system.

In our response above, we note that lipids will bind to membrane proteins at specific site(s) and weaker sites, often described as non-annular lipids. The revision includes this discussion point.

1. If a single mutation is introduced to the inner cavity, its effect will be "doubled" because the inner cavity is shared by two identical subunits. This effect needs to be clarified in the result section.

Great point. In addition, an outer mutant will also impact not one but both outer binding site(s)s. The revised manuscript makes note of this point.

1. In the result section, "Mutant cycle analysis of KDL binding to vanadate-trapped MsbA.":4a. It seems necessary to show the mass spectra for Msb-ADP-vanadate complex as well as its lipid bound forms.

In the original submission, the mass spectra of vanadate trapped MsbA with KDL binding was provided in Supplementary Figures 10 and 11.

4b. The rationale of this section (i.e., what mechanistic insights can be obtained from this study) is unclear. For example, it is not sure what meaningful information can be obtained from a single type (ADP/vanadate) of the bound state regarding the ATP-driven function of MsbA.

MsbA is a dynamic, populates different conformations. Trapping with vanadate locks the transporter in an outwardfacing state with NDB interacting. This provides the opportunity to characterize binding to the exterior site. We revised the manuscript to note this point.

**Reviewer #3 (Public Review):**
Summary:In this paper presented by Liu et al, native MS on the lipid A transporter MsbA was used to obtain thermodynamic insight into protein-lipid interactions. By performing the analyses at different lipid A concentrations and temperatures, dissociation constants for 2-3 lipid A binding sites were determined, as well as enthalpies were calculated using nonlinear van't Hoff fitting. Changes in free Gibb's energies were then calculated based on the determined dissociation constants, and together with the enthalpy values obtained via van' t Hoff analysis, the entropic contribution to lipid binding (DeltaS*T) was indirectly determined.Strengths:This is an extensive high quality native MS dataset that provides unique opportunities to gain insights into the thermodynamic parameters underlying lipid A binding. In addition, it provides coupling energies between mutations introduced into MsbA, that are implicated in lipid A binding.Weaknesses:The data all rely on the accuracy of determining KD values for lipid binding to MsbA. For the weaker binding sites, the range of lipid concentrations probed were in fact too low to generate highly accurate data. Another weakness is a lack of clear evidence, which KD values belong to which of the possible lipid A binding sites.

See our detailed response to reviewer 1 regarding Kd determination using native MS compared to other techniques. We chose to focus on the first three lipid binding events and adjusted the concentrations accordingly to titrate these three. As noted above, the Kd values can be determined from one mass spectrum. For rigor, we include different titration points and fit sequential binding model to the data – the fits are shown in supplemental and quite reasonable.

Regarding multiple lipids binding to different site(s), we have been able to distinguish high-affinity vs low-affinity PIP binding to Kir3.2 in a previous study [4]. This was apparent by the mole fraction curves for some lipid bound states not returning back to zero. We agree binding to multiple sites can be an issue. However, other techniques report on the ensemble of binding and, hence, no real useful information is obtained. Native MS enables one step in the right direction by dissecting the different lipid bound states. Future directions will need to further address this forefront question in the field, which we make point of now in discussion.

**Reviewer #1 (Recommendations For The Authors):**
Experiments/analysis: In short, there should be a proof of principle experiment that the thermodynamic constants determined by MS are accurate. Once that is done, the authors can add a more engaging structural interpretation of the results from the mutant cycles (which they seem to consciously avoid in the present manuscript?). How are cooperative residues coupled? Why?

See our detailed response to reviewer 1 above.

The manuscript is well-written, but Figures 3-5 are somewhat repetitive and require a lot of time to understand. Schematics of the main findings in each figure would help the uninitiated reader.

We agree the illustrations are complex but there is rich data being shown.

Figure 2 C contains an x-axis label error.

Corrected.

**Reviewer #2 (Recommendations For The Authors):**
1. Lines 128-129: "Like other mutant cycle studies, we assume the single- and double-mutations do not disrupt binding at specific sites on MsbA."This statement is obscure and needs to be clarified. Does this mean that the mutations still allow binding of KDL, or the mutations do not disrupt the conformational integrity of the binding sites?

This statement has been removed.

1. Lines 137-139: "More specifically, R78 coordinates one of the characteristic phosphoglucosamine (P-GlcN) substituents of KDL whereas K299 interacts with a carboxylic acid group in the headgroup of KDL."Two identical subunits form a dimer interface that forms an LPS binding site. Thus, a single mutation on the inner cavity will disrupt two binding sites on LPS. One R78 to P-ClcN and the other to a sugar backbone. Also, one K299 interacts with a carboxylic acid group in the headgroup and the other to an unknown (not clear in the figure).

Also noted above, mutation of the outer site will also impact the two outer sites. We have made note of this caveat.

1. Lines 171-172: "leading to an increase in ΔG by ~4 kJ/mol (Fig. 2d)"Relative to what?

Corrected.

1. Lines 172-173: "Mutant cycle analysis indicates a coupling energy (ΔΔGint) of 1.7 (plus minus) 0.4 kJ/mol that contributes to the stability of KDL-MsbA complex."The sign of DDG (DDH,DDS)_int is a bit confusing. I recommend that authors define the meaning of negative or positive sign of DDG_int (DDH,DDS) at this point. Here, a positive sign means favorable cooperation between the two mutated residues. Sometimes, researchers designate a positive cooperativity as a negative sign.

The literature on mutant cycles does not appear to follow a consensus on the sign. Here, we have revised the manuscript to note positive sign means favorable cooperation and follow the formalism recently described by Horovitz, Sharon, and co-workers [5].

1. Lines 182-185: "Enthalpy and entropy for KDL binding MsbA R188A was largely similar to the wild-type protein (Fig 3a). However, the R243A mutation resulted in an increase in entropy, compensated for by an increase in positive enthalpy (Fig 3a)."The thermodynamic parameters for R243A mutation change in a similar manner to WT and R188A. It is R238A, not R243A, whose DH-DS interplay shows a distinct pattern from WT. Please, reword this sentence.

The sentence has been revised.

1. Lines 252-253: Solvation of polar groups in aqueous solvent has been ascribed to positive heat capacities whereas negative for apolar solvation.This statement is not precise. More precisely, the collapse of apolar molecules from their solvated state leads to the negative "change" in heat capacity.

The sentence has been corrected.

1. Line 262-267: "These hydrophilic patches will be highly solvated, which will be desolvated upon binding lipids contributing favorably to entropy. In the case of MsbA, the selected lysine and arginine residues (based alpha carbon position) are separated by about 9 to 18 Å (PDB 8DMM). This distance could result in overlap of solvation shells that collectively contribute to the positive coupling enthalpy observed for MsbA-KDL interactions."This statement is too speculative without presenting the degree of solvation of the residues targeted for mutation. More quantitative arguments seem to be needed.

We have removed the speculative statement.

**Reviewer #3 (Recommendations For The Authors):**
In this paper presented by Liu et al, native MS on the lipid A transporter MsbA was used to obtain thermodynamic insight into protein-lipid interactions. By performing the analyses at different lipid A concentrations and temperatures, dissociation constants for 2-3 lipid A binding sites were determined, as well as enthalpies were calculated using nonlinear van't Hoff fitting.Changes in free Gibb's energies were then calculated based on the determined dissociation constants, and together with the enthalpy values obtained via van' t Hoff analysis the entropic contribution to lipid binding (DeltaS*T) was indirectly determined.

Correction – In the case on linear van’t Hoff plots, dH and dS were determined directly from the plot. For the nonlinear form of the van’t Hoff equation, which does not include an entropy fitting parameter, we back calculated dS using dH and dG at a given temperature.

The authors then included single, double and triple mutants of residues known based on cryo-EM and X-ray structures to interact with Lipid A either in the large inward-facing cavity or at a secondary binding site accessible at the surface of outward-facing MsbA, and determined the thermodynamic parameters of these mutants alone and combined to gain access to coupling energies of pairwise interactions. This method has its roots in studying pair-wise interactions of protein-protein interfaces, generally known as thermodynamic mutant cycle analysis.Having the main expertise in ABC transporter structure-function, I will judge the paper mostly from the standpoint of what I can learn as a transporter expert from this study and whether the insights are of value for researchers with average biophysical knowledge.My overall impression of the manuscript is that, while it contains a wealth of experimental data using the innovative and unique method of native mass spectrometry, it is hard to understand what one can learn from this analysis beyond their interesting key finding that entropy plays an important role in lipid binding (but only at certain temperatures). In particular, the lessons learned from the coupling energy analysis of the introduced mutations is hard to grasp/digest for me with regards to what I can learn from these numbers (other than learning that there are such coupling effects).

We agree the thermodynamic data is rich. Often a ddGint of zero is reported as having no coupling/significance but here the value is due to compensating ddH and d-dTS terms. In our view, this work forms the foundation of additional studies to better understand the coupling energetic terms, beyond ddGint.

In some instances, the text/figure legends are a bit unclear or contain some typos; but this part can easily be handled in a revision. The discussion is well written and embeds the main findings in the (still rather limited) literature on thermodynamic analyses of lipid binding of membrane proteins.Major points1. The authors may have clarified the following point in a previous paper; but at least in this paper, it is unclear to me how they purified MsbA without lipid A. The reason I am asking is that in our experience, if one purifies MsbA expressed from *E. coli* with standard detergents (e.g. beta-DDM) one will find a perfect density for Lipid A when determining an inward-facing structure by cryo-EM. According to the Methods, MsbA is purified initially in DDM, and rebuffered to C10E5 during size exclusion chromatography. When looking at Fig. 2b, the authors state (or assume?) that if no lipid A is added, MsbA has 0 % lipid A bound.

We have previously reported details of MsbA sample prep and optimization [6]. The revised manuscript makes note of this previous work and refers the reader to the publication. Yes, we see no appreciable signal for lipid A bound to MsbA (see Fig 2b).

We also note that samples of MsbA prepared using DDM is highly heterogenous, contaminated by a battery of small molecules (that we suspect are co-purified lipids). These contaminants will inadvertently impact biochemical studies.

1. A second topic where further clarification is in my view needed is the question of the conformations that were probed and the lipid binding sites. If I get the experimental rationale correctly, most of the data were determined in the absence of nucleotides, and only a small subset (Fig. 5) of data were determined in the presence of ATP-vanadate. However, structural evidence for the cytosolic lipid A binding site has been only determined for outward-facing MsbA (PDB: 8DMM), but has thus far not been seen in any of the inward-facing cryo-EM structures of MsbA, including recent well-resolved cryo-EM structures showing excellent density for the lipid A bound to the inward-facing cavity (PDB: 7PH2). Further, there is only one lipid A molecule that can be accommodated by the inward-facing cavity, whereas (owing to the symmetry of the homodimer) two lipid A can be bound sideways to outward-facing MsbA. Now, my understanding problem is why one does see up to three lipid A molecules bound to inward-facing apo MsbA, e.g. Fig. 2b and elsewhere. Where are they expected to bind? And what is the evidence supporting these additional binding sites?

See our detailed response to reviewer 1. If we add more lipid, we see more lipid binding to MsbA, like every other membrane protein we have studied. This data clearly indicates that there are more KDL binding site(s) – deciphering the affinity of these site(s) represents a problem on the horizon.

A further question is which lipid A binding sites are present in vanadate-trapped MsbA. Here, there are two identical binding sites (at the surface of each MsbA molecule), and it is therefore surprising to see that the affinities for the first and the second binding site are so different (see e.g. Supplementary Fig. 13).

Great point. A logical explanation (described for other biochemical systems) is the two exterior LPS binding sites display negative cooperativity i.e., binding at one site weakens the affinity at the other site.

Finally, what is the evidence that in vanadate-trapped MsbA, all molecules have closed NBDs and thus assume the outward-facing conformation? It is not uncommon that vanadate trapping leads to NBD closure only in a subfraction of all transporters (hence not in 100 % of them).

Yes, the native mass spectrum shows no appreciable signal for MsbA not trapped with vanadate/ADP. In our previous cryoEM study [6], using the vanadate-trapped transporter, we did not observe particles with NDBs dissociated in space. Regarding samples from other labs, a native mass spectrum could shed light into the population of untrapped protein – however, most studies use SDS-PAGE for quality control of their purified samples. This technology is not sufficient to address underlying biochemical issues.

We do have a new report in preparation describing a new discovery regarding trapping efficiency of MsbA.

1. The key parameter that is underlying the entire thermodynamic analysis of wt and mutant MsbA is the dissociation/association constant, which are used to calculate free Gibb's energy and, via van't Hoff analysis, enthalpy. Entropy is not determined directly, but in fact indirectly from these two numbers both depending on the measurement quality of dissociation/association constant. Now, when looking at the fitted curves as shown in Figure 2b (and in the supplement), determination of the dissociation constant for KDL1 (blue curves) look reasonable and the determined KDs are within the range of measured points. However, for KDL2 (red) and even more so KDL3 (yellow), the determined KD values (Supplementary Table 5), the measured KD values are typically higher than highest KDL conc used in the assay (1.5 uM). For this reason, and despite the fact that error bars of the fits look reasonably small, I still have doubts about the reliability of these KD values for KDL2 and KDL3.Hence, the surprisingly strong changes of enthalpy/entropy values for different mutants/temperatures may have their origin in incorrectly determined KD values.

The increase in binding affinity of subsequent lipid binding events is consistent with many reports from our group [1, 2, 4, 6-9] and that of Prof. Robinson [10, 11] on this topic. As noted above, we indeed observe linear van’t Hoff plots with positive and negative slopes as well as non-linear curves that are convex or concave. The MsbA protein (wt or mutant), trapped or not, all display unique temperature-dependent responses. If the reviewer suggestion that the Kd values are incorrectly or randomly determined, then none of the binding data should follow thermodynamic van’t Hoff equations. This is simply not the case - the error bars and fits are reasonable. Backing up even further, looking the raw native mass spectra (see supplemental figure 1-3 and 10-11) one can see different temperature-dependence of lipid binding.

Minor points1. Lines 116-131: this section reads as an extended introduction/aims, and does not contain any results.

This section has been moved to introduction.

1. Lines 137-139: suggested to check whether these interactions are also present in recently determined cryo-EM structures determined at fairly high resolution (PDB: 7PH2)

The interactions of MsbA and LPS (bound at the interior site) are comparable for PDB 7PH2 and 6BPL.

1. Lines 144-146: suggested to elude in more detail on the fitting procedure here, as the KD values determined in this way are the foundation of all quantitative assessments.

Details of data analysis and the fitting procedure are provided in methods.

1. Figure legend, Fig. 2: Technically, MsbA was solubilized and purified in DDM and detergent exchange was done on SEC to C10E5.

Corrected.

1. Figure legend, Fig. 4: description in (a) on deconvoluted mass spec data is incorrect. Letter below needs to be adjusted accordingly.

Corrected.

1. Figure legend, Fig. 5: suggested to mention in Figure legend title that here we look at ADP-vanadate trapped MsbA.

Corrected.

References

1. Cong, X., et al., Determining Membrane Protein–Lipid Binding Thermodynamics Using Native Mass Spectrometry. Journal of the American Chemical Society, 2016. 138(13): p. 4346-4349.

2. Cong, X., et al., Allosteric modulation of protein-protein interactions by individual lipid binding events. Nat Commun, 2017. 8(1): p. 2203.

3. Qiao, P., et al., Insight into the Selectivity of Kir3.2 toward Phosphatidylinositides. Biochemistry, 2020. 59(22): p. 2089-2099.

4. Qiao, P., et al., Entropy in the Molecular Recognition of Membrane Protein-Lipid Interactions. J Phys Chem Lett, 2021. 12(51): p. 12218-12224.

5. Sokolovski, M., et al., Measuring inter-protein pairwise interaction energies from a single native mass spectrum by double-mutant cycle analysis. Nat Commun, 2017. 8(1): p. 212.

6. Lyu, J., et al., Structural basis for lipid and copper regulation of the ABC transporter MsbA. Nat Commun, 2022. 13(1): p. 7291.

7. Patrick, J.W., et al., Allostery revealed within lipid binding events to membrane proteins. Proc Natl Acad Sci U S A, 2018. 115(12): p. 2976-2981.

8. Schrecke, S., et al., Selective regulation of human TRAAK channels by biologically active phospholipids. Nature Chemical Biology, 2021. 17(1): p. 89-95.

9. Zhu, Y., et al., Cupric Ions Selectively Modulate TRAAK-Phosphatidylserine Interactions. J Am Chem Soc, 2022. 144(16): p. 7048-7053.

10. Tang, H., et al., The solute carrier SPNS2 recruits PI(4,5)P(2) to synergistically regulate transport of sphingosine1-phosphate. Mol Cell, 2023. 83(15): p. 2739-2752 e5.

11. Yen, H.Y., et al., PtdIns(4,5)P(2) stabilizes active states of GPCRs and enhances selectivity of G-protein coupling. Nature, 2018. 559(7714): p. 423-427.